# PALOMA : A Benchmark for Evaluating Language Model Fit

**Ian Magnusson**♠    **Akshita Bhagia**♠    **Valentin Hofmann**♠    **Luca Soldaini**♠
**Ananya Harsh Jha**♠    **Oyvind Tafjord**♠    **Dustin Schwenk**♠    **Evan Pete Walsh**♠
**Yanai Elazar**♠◇    **Kyle Lo**♠    **Dirk Groeneveld**♠    **Iz Beltagy**♠    **Hannaneh Hajishirzi**♠◇
**Noah A. Smith**♠◇    **Kyle Richardson**♠    **Jesse Dodge**♠
♠Allen Institute for Artificial Intelligence
◇Paul G. Allen School of Computer Science & Engineering, University of Washington
{ianm,jessed}@allenai.org

## Abstract

Evaluations of language models (LMs) commonly report perplexity on monolithic data held out from training. Implicitly or explicitly, this data is composed of domains—varying distributions of language. We introduce PERPLEXITY ANALYSIS FOR LANGUAGE MODEL ASSESSMENT (PALOMA)[1], a benchmark to measure LM fit to 546 English and code domains, instead of assuming perplexity on one distribution extrapolates to others. We include two new datasets of the top 100 subreddits (e.g., *r/depression* on Reddit) and programming languages (e.g., *Java* on GitHub), both sources common in contemporary LMs. With our benchmark, we release 6 baseline 1B LMs carefully controlled to provide fair comparisons about which pretraining corpus is best and code for others to apply those controls to their own experiments. Our case studies demonstrate how the fine-grained results from PALOMA surface findings such as that models pretrained without data beyond Common Crawl exhibit anomalous gaps in LM fit to many domains or that loss is dominated by the most frequently occurring strings in the vocabulary.

## 1 Introduction

Progress in AI is catalyzed by evaluations that define new ways of measuring progress (Deng et al., 2009, Wang et al., 2018, and Wang et al., 2019, *inter alia*). Language models (LMs) often evaluate LM fit as loss or perplexity [Jelinek et al., 1977] on held out training data or few traditional test sets (Chelba et al., 2013, Merity et al., 2016, *inter alia*). These loss measures have been shown to improve predictably with increases in training compute [Kaplan et al., 2020, Hoffmann et al., 2022] and loss may predict performance on downstream tasks [Xia et al., 2022, Gadre et al., 2024, Du et al., 2024]. However, scaling pretraining data aggregates more domains that LMs implicitly learn to model [Diaz and Madaio, 2023, Aharoni and Goldberg, 2020]. Does rising performance lift all data? Or do some domains capture most improvement in LM fit? How do we evaluate what language distributions models learn from different pretraining data? What domains should studies evaluate loss on to measure the relationship of loss and downstream performance? To answer these questions, perplexity evaluations ought to measure LM fit to many domains, rather than extrapolating trends from a single prescriptive mix of domains.

---

[1]Dataset and links to code repository are available at `https://paloma.allen.ai`

38th Conference on Neural Information Processing Systems (NeurIPS 2024) Track on Datasets and Benchmarks.

In this work we introduce PALOMA, a benchmark to study LM fit on many domains. We measure perplexity on different distributions of language sampled from 16 sources, such as C4 [Raffel et al., 2019], that have metadata such as URLs marking 546 textual domains. Beyond evaluation data, we aim to enable and enrich fair comparisons for scientific research on language modeling with the following artifacts: guidelines for comparing LM fit, 6 baseline 1B parameter models pretrained on popular corpora, and standardized code for experiments with PALOMA.

As reproducing pretrained models for every new project is onerous, we provide standard training controls for benchmark decontamination and training data order to orchestrate a greater density of comparisons across the research community. We also control how PALOMA is evaluated by fixing sample size per domain, model vocabulary, and inference format. Lastly, we demonstrate how to make fair comparisons over two measures of cost, number of model parameters and training tokens, enabling assessment of hardware-agnostic efficiency and the measurement of scaling trends.

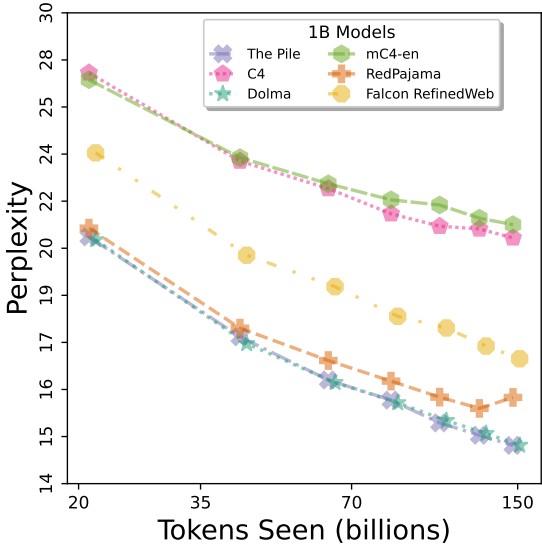

Figure 1: Perplexity on PALOMA for baselines pretrained with our experimental controls such as benchmark decontamination. We measure fit over diverse sources beyond data held-out from training. PALOMA enables loss comparisons between *different* models, such as this figure where pretraining data is varied while all other factors are controlled. This measurement excludes documents from fringe sources and code data not supported by our decontamination approach.

Among the 16 sources curated in our benchmark, we contribute two new datasets constructed from data held out of DOLMA [Soldaini et al., 2024]: (1) a subsample of the top 100 subreddits by number of comments, and (2) code from the top 100 programming languages by number of tokens. Also, we repurpose corpora of fringe online communities to measure LM fit to discourse previously studied for the prevalence of toxicity and hate speech [Ribeiro et al., 2021, Zannettou et al., 2018, Papasavva et al., 2020]. While, capturing domains required by all possible lines of research is impossible for any one benchmark, PALOMA focuses on English and code data and aims to assemble the most fine-grained domains readily identifiable from existing metadata.

To demonstrate possible uses of results from our dataset, we present a series of case studies in §4. Among other findings, our experiments isolate change in fit from which pretraining corpus is used (Figure 1) and find that pretraining without heterogeneous data sources beyond Common Crawl can lead to perplexities in some domains that do not improve consistently with number of tokens seen. We also find that few vocabulary types account for most of the loss measured in perplexity.

In sum, PALOMA contributes:

1. Curated release of the most fine-grained perplexity evaluation data in use in LM research, along with guidelines and code for standardized and rigorous perplexity evaluation.

2. New evaluation data for the 100 most popular subreddits and programming languages.

3. 1B LMs pretrained on C4, MC4-EN, FALCON REFINEDWEB, THE PILE, REDPAJAMA, and DOLMA with controlled hyperparameters, token budget, benchmark decontamination, and training order for fair comparisons, along with code for others to do the same.

4. Case studies demonstrating analyses that are possible with PALOMA, such as finding that pretraining without data beyond Common Crawl leads to inconsistent fit to many domains and that perplexity is driven by improved fit on the most common vocabulary strings.

## 2  Sources of evaluation data

| Purpose | Source | Val. + Test Tokens | Domains | Tokens per Split per Domain |
|---|---|---|---|---|
| Standard language modeling benchmarks | C4 [Raffel et al., 2019] | 2,000,000 | 1 | 1,000,000 |
| | MC4-EN [Chung et al., 2023] | 2,000,000 | 1 | 1,000,000 |
| | WIKITEXT-103 [Merity et al., 2016] | 531,103 | 1 | 265,552 |
| | PENN TREEBANK [Marcus et al., 1999] | 191,735 | 1 | 95,868 |
| | REDPAJAMA [Together Computer, 2023] | 1,399,946 | 7 | 99,996 |
| | FALCON REFINEDWEB [Penedo et al., 2023] | 2,000,000 | 1 | 1,000,000 |
| | DOLMA [Soldaini et al., 2024] | 5,994,901 | 6 | 499,575 |
| Fine-grained domain benchmarks | M2D2 S2ORC [Reid et al., 2022] | 33,374,351 | 167 | 99,923 |
| | M2D2 WIKIPEDIA [Reid et al., 2022] | 9,780,719 | 49 | 99,803 |
| | C4-100-DOMAINS [Chronopoulou et al., 2022] | 19,609,392 | 99 | 99,037 |
| | DOLMA-100-SUBREDDITS [Soldaini et al., 2024] | 19,360,263 | 100 | 96,801 |
| | DOLMA-100-PROGRAMMING-LANGUAGES [Soldaini et al., 2024] | 19,999,613 | 100 | 99,998 |
| Disparities | TWITTERAAE [Blodgett et al., 2016] | 1,441,263 | 2 | 360,316 |
| Fringe sources | MANOSPHERE CORPUS [Ribeiro et al., 2021] | 1,999,915 | 9 | 111,106 |
| | GAB CORPUS [Zannettou et al., 2018] | 2,000,000 | 1 | 1,000,000 |
| | 4CHAN CORPUS [Papasavva et al., 2020] | 2,000,000 | 1 | 1,000,000 |
| | **PALOMA** | 123,683,201 | 546 | 113,263 |

Table 1: The 16 data sources sampled to create language modeling evaluations in PALOMA (§2), organized by the purpose for inclusion. These coarse-grained sources contain finer-grained domains, which use metadata to distinguish distinctive distributions of language such as a subreddit for discussing board games. PALOMA aims to enable research on differences in LM fit over hundreds of domains by curating and standardizing the text datasets with the most fine-grained domains readily available from existing metadata. We target a minimum of 100 thousand tokens per domain and 1 million tokens per source to select a balance between inference cost and metric variance.

We define two terms: *Sources* are as existing datasets (or curated subsets there of) in use for research. *Domains* are fine-grained partitions of sources based on available metadata that attempt to surface a distinct and intuitive distribution of language (e.g., Wikipedia articles about visual arts or a subreddit for advice on PC builds). PALOMA is derived from 16 sources further divided into 546 domains (see Table 1).[2] Where we curate previous fine-grained corpora, we inherit their operationalization of domains, ranging from the community-driven Wikipedia ontology to expert curation and automatic classification. Where we build our own fine-grained domains from Reddit and GitHub, we make similar use of metadata about subreddits and file extensions.

Compared to monitoring monolithic validation loss during model development, interpreting LM fit to specific fine-grained domains poses unique challenges. Crucially, we must not assume better LM fit to a domain reflects improvements in the specific skills that are valued by the humans producing language in that domain [Diaz and Madaio, 2023]. For instance, we might expect overlapping domains for academic papers in both DOLMA and REDPAJAMA to exhibit similar perplexities for a given model, perhaps assuming perplexity represents how much a model captures knowledge about relevant academic fields. But domains can also differ due to preprocessing when texts were collected in each source rather than from how texts were composed by their original authors. So instead of relying on LM fit to measure what we think a model *should* learn about a domain, we examine anomalies in domain fit to see what a model *is* learning. We find that the same model can have 391,171 perplexity on arXiv in REDPAJAMA and 14 on the overlapping academic domain, peS2o, in DOLMA (§4.1). In this approach we follow Holtzman et al. [2023] and McCoy et al. [2023] by aiming to examine model behaviors, regardless of their desirability to humans.

Also note that PALOMA focuses on English and code data, as most current LMs also emphasize these types of data. However, we strongly encourage future work to explore fit to fine-grained domains in other languages.

The rest of this section addresses each source, why we include it, and how it identifies any domains it contains (all 546 domains are listed in Appendix E).

---

[2]Unless stated, token counts are computed with the GPT-NeoX-20B tokenizer [Black et al., 2022].

**Standard language modeling sources**   Though it is common practice to evaluate on held out data from the pretraining corpus of a given model, we evaluate *across* several standard corpora. **C4** [Raffel et al., 2019, Dodge et al., 2021] and **MC4-EN** [Chung et al., 2023] are language model training datasets created by taking the snapshots of Common Crawl data and applying a number of filters with the intention of retaining "high-quality", natural language. Both datasets are filtered to retain natural English, and in this work we only use the English portion of MC4-EN. **WIKITEXT-103** [Merity et al., 2016] and **PENN TREEBANK** [Marcus et al., 1999] are classic datasets that have been used to evaluate language model perplexity for decades (Radford et al., 2019, Brown et al., 2020, Rae et al., 2021, Hoffmann et al., 2022, *inter alia)*. WIKITEXT-103 is text from Wikipedia articles, and PENN TREEBANK [Marcus et al., 1999] is a set of 1989 Wall Street Journal articles[3]. **REDPAJAMA** [Together Computer, 2023] is an attempt at reproducing the data mixture from LLaMA [Touvron et al., 2023] from sources such as webtext, Wikipedia, arXiv, and StackExchange. It was used to train RedPajama-INCITE [Together Computer, 2023]. **FALCON REFINEDWEB** Penedo et al. [2023] was created from all Common Crawl scrapes until June 2023 by applying relatively interpretable filters, and is a subset of the Falcon models' training data [Almazrouei et al., 2023]. **DOLMA** Soldaini et al. [2024] is made of Common Crawl, Wikipedia, books, academic papers, code repositories, and Reddit, and was used to train OLMo models [Groeneveld et al., 2024].

**Fine-grained domain sources**   We include datasets with the most fine-grained metadata marking hundreds of domains. **M2D2** [Reid et al., 2022] is made of academic papers from S2ORC [Lo et al., 2020] and text from Wikipedia, organized into a two-level hierarchy by academic field categories or Wikipedia ontology, respectively. We sample both top-level domains and lower-level subdomains. **C4-100-DOMAINS** [Chronopoulou et al., 2022] is text from the 100 internet domains with the most pages in C4.[4] **DOLMA-100-SUBREDDITS** and **DOLMA-100-PROGRAMMING-LANGUAGES** are two evaluation sets we introduce in this work sampled from DOLMA [Soldaini et al., 2024]: the former is text from the top 100 subreddits (ranked by number of posts), and the latter is the top 100 programming languages by number of tokens in the THE STACK [Kocetkov et al., 2022]. See Appendix E for more details.

**Disparities between speech communities**   LMs today primarily process dominant dialects in countries, such as the US, where they are most often trained and deployed. Even within English, hundreds of millions of people around the world speak other dialects that have been shown to be underserved by existing models [Blodgett et al., 2016]. As a starting point for measuring disparities between dialects, we include **TWITTERAAE** [Blodgett et al., 2016], two corpora representing African-American and White-aligned English, automatically classified via geolocation information and demographic census statistics. [5]

**Fringe sources previously studied for problematic discourse**   LM fit to these fringe texts characterizes model exposure to distinct social contexts in which toxic language arises. **MANOSPHERE** [Ribeiro et al., 2021], **GAB** [Zannettou et al., 2018], and **4CHAN CORPORA** [Papasavva et al., 2020] are three fringe corpora which contain larger proportions of hate speech and toxicity than mainstream sources like Wikipedia or Twitter. These texts span 2006-2019 and include independent message boards and subreddits sharing a masculinist ideology, Gab (an alt-right focused Twitter alternative with minimal moderation), and the Politically Incorrect board (/pol/) of 4chan, a fringe imageboard emphasizing anonymity and ephemerality.

---

[3]PENN TREEBANK is pretokenized, and uncommon words are replaced with a special "unknown" token.

[4]Four of the 100 domains have less than the 100 thousand tokens per split that we aim for.

[5]We follow the reproduction of this dataset used in HELM [Liang et al., 2022], but we fix an error in loading escaped sequences of the data that, among other issues, renders emojis as literal hexadecimal bytes.

# 3 Perplexity evaluations done right

**Guidelines** Fairly evaluating different models using perplexity is hard. To do so, we must account for factors that can confound results with guidelines for training (G1, G2) and evaluation (G3, G4, G5).

G1 DECONTAMINATION: Remove *pretraining* data that leaks evaluation data to ensure validity of perplexity evaluation.

G2 TRAINING ORDER: Where possible, keep the training data order the same to control differences from recency effects.

G3 SUBSAMPLING: Subsample size poses a tradeoff between inference cost and variance. Size subsamples to tolerate variance equally for each domain.

G4 VOCABULARY: Vocabulary determines the event space of possible sequences and the comparability of perplexity measurements. Normalizing likelihood by a segmentation intrinsic to the text (e.g., bytes) partially addresses this, but fixing the vocabulary is preferable.

G5 EVALUATION FORMAT: Use a consistent implementation of perplexity to ensure comparability regarding engineering details such as the handling maximum sequence lengths.

**Experimental controls** Our code repository[6] releases controls that implement each guideline. Here we briefly explain each (complete specification of our experimental controls is provided in Appendix C).

For G1, we use a Bloom filter [Bloom, 1970] to detect exact match overlaps of pretraining and evaluation data. We match text at the paragraph level, i.e., newline separated spans of text. To avoid coincidental collisions in the space of small strings, we ignore matches in paragraphs smaller than 13 unicode segmented tokens [Unicode, 2023]. Similarly, we ignore paragraphs composed of only punctuation, spaces, and emoji. Lastly, as code data consists almost entirely of short and often repeated lines, we forgo any decontamination on these sources (DOLMA-100-PROGRAMMING-LANGUAGES and the THE STACK domain of DOLMA). Finally, we remove whole pretraining documents if they contain *any* contaminated paragraph.

For G2, contemporary LMs train on instances that are maximum sequence length concatenations of training documents, so we must fix the order of concatenated instances. We achieve this by fixing the tokenization, maximum sequence length, and random seed, as well as providing dataloading code where order is invariant to number of devices.

For G3, we empirically observe how variance in perplexity over subsamples of C4 evaluation data grows inversely to sample size (Appendix C.2.1). Extrapolating from these results to select desired thresholds for variance, we pick 1 million and 100 thousand tokens as our target size for sources and domains, respectively.

For G4, where possible we fix model vocabulary to GPT-NeoX-20B's [Black et al., 2022] with 3 special tokens added by Groeneveld et al. [2024]. When vocabulary must be changed, for instance comparing to off-the-shelf models, we follow THE PILE [Gao et al., 2020] and use bits per byte (BPB; Appendix B).

For G5, we follow the input format established by THE PILE [Gao et al., 2020]. This format evaluates documents individually, rather than packed into concatenated maximum sequence length inputs. Documents longer than maximum sequence length are split into disjoint inputs.

In Table 2 we compare how PALOMA implements controls for these guidelines against practices in previous LM benchmarks. PALOMA is the first benchmark to remove contamination across all pretraining data. THE PILE [Gao et al., 2020] note that they only address decontamination partially by deduplicating 2 of 22 domains at the document level before splitting. PALOMA is also the first

---

[6]https://github.com/allenai/OLMo-Eval/tree/main/paloma

| Guideline | THE PILE [Gao et al., 2020] | M2D2 [Reid et al., 2022] | C4-100-DOMAINS [Chronopoulou et al., 2022] | HELM LM Scenarios [Liang et al., 2022] | PALOMA |
|---|---|---|---|---|---|
| G1 DECONTAMINATION | partial, doc-level | none | none | not required | sub-doc-level |
| G2 TRAINING ORDER | not required | not required | not required | not required | fixed |
| G3 SUBSAMPLING | uniform | uniform | uniform | inherits splits | stratified |
| G4 VOCABULARY | not required | not required | not required | not required | fixed |
| G5 EVALUATION FORMAT | no concat or overlap | not required | not required | API dependent | no concat or overlap |
| # Domains | 22 | 216 | 99 | 14 | 546 |

Table 2: Differences between PALOMA and other language modeling benchmarks on guidelines (§3) for experiments of assessing LM fit. Ours is the first perplexity benchmark to remove contaminated training data, fix training order, sample domains equally, and fix vocabulary. We also adopt a controlled inference format from Gao et al. [2020].

contemporary perplexity benchmark to recommend and implement a method to fix the training data order, to apply stratified sampling to evaluation domains, and to recommend fixing vocabulary. THE PILE and HELM also detail their evaluation formats, but we note that HELM's inference code depends on calls to proprietary APIs which may not remain reproducible for some models.

**Comparability**   When using PALOMA to compare models, we recommend that researchers also adopt our experimental controls or note as a limitation to comparability any uncontrolled factors. We also recommend that measures of cost are considered when comparing models on PALOMA, specifically number of model parameters and number of tokens seen in training. Complimentary to work that focuses on realized costs such as energy use, FLOPs, or GPU hours [Peng et al., 2023], we elect to measure these more abstract cost values so that our efficiency comparisons are agnostic to hardware. Finally, as LMs trained with non-constant learning rate schedules scale sub-optimally until improving when learning rate drops towards the end of training, fair comparisons involving intermediate checkpoints should be matched with respect to the portion of total optimization steps completed.

By providing fair comparisons, the following types of claims about perplexity performance can be made with our benchmark: (1) which among compute-matched models performs best, (2) which models reach a given performance with the least compute, (3) which pretraining corpus produces models with best performance, (4) quantifying the trend of performance as a function of scale.

**Metric**   PALOMA uses standardized inference code to compute metrics to assess LM fit to the evaluation data we have curated. Perplexity [Jelinek et al., 1977] is our primary metric (others not used in the body of this paper are detailed in Appendix B). Unless otherwise stated, we use perplexity to mean perplexity per token, where a log likelihood $\ell$ over documents $N = \{t^1, \ldots, t^{|N|}\}$ is normalized by $\mathbf{T}(N)$ denoting the number of tokens in the documents (i.e., $\mathbf{T}(N) = \sum_{t \in N} |\mathbf{tokenize}(t)|$):

$$\ell = \sum_{t \in N} \sum_{i}^{|t|} \ln p(t_i \mid t_{<i})$$

$$\text{perplexity} = e^{-\frac{\ell}{\mathbf{T}(N)}}$$

## 4   Case studies

In this section, we present one full case study and a single conclusion from a second. In Appendix D we present additional studies, demonstrating the types of analyses possible with PALOMA.

## 4.1 Pretraining Beyond Common Crawl Shows Improved Stability of LM Fit

We hypothesize that one of the strongest drivers of differences in performance between different domains is the composition of the pretraining data of a language model. While we show in Appendix D.1 that scaling model parameters or tokens seen increases performance on nearly all domains, the pretraining data composition directly determines the distribution of language that the model is learning to fit, which may or may not align with the distributions of language in the domains we evaluate. Therefore we examine the impact of varying the pretraining corpus while holding all other experimental decisions the same.

**Baseline Models** We train and release a set of 6 baseline models on common pretraining corpora following our training guidelines (§3). Training these models ourselves allows us to apply decontamination and fixed order to their pretraining data as well as using a standard tokenizer to enable the greatest level of comparability. These models are 1B parameter models trained for ∼150B tokens on DOLMA [Soldaini et al., 2024], THE PILE [Gao et al., 2020], REDPAJAMA [Together Computer, 2023], FALCON REFINEDWEB [Penedo et al., 2023], C4 [Raffel et al., 2019, Dodge et al., 2021], and MC4-EN [Chung et al., 2023]. Additional training details are included in Appendix G.

**Ordinary perplexity** In Figure 1, we consider the most simple and aggregated view of LM fit that PALOMA can provide—perplexity as defined in §3. Specifically we compute perplexity over all data, excluding the three fringe sources with prevalent toxicity. We also exclude code data in DOLMA and DOLMA-100-PROGRAMMING-LANGUAGES.[7]

Using this view, we see that baseline models trained only on Common Crawl data (C4, FALCON REFINEDWEB, and MC4-EN) stand out from the others which incorporate more curated data sources. However, this points to the limitation of this most aggregated view of the results: ordinary perplexity represents fit to domains in proportion to the number of tokens we have chosen to sample from each domain. We sample 100,000 tokens from each domain and the majority of our domains are not sourced from Common Crawl. So Common Crawl is much less represented in PALOMA than in most pretraining corpora, which typically consist of mostly Common Crawl as this is the most abundant public source of text data. Nevertheless this simplified view of the results is useful for specific use cases that need a single metric over a prescriptive mix that emphasizes robustness to a diversity of domains, largely derived from non-web scraped sources.

**Macro average perplexity** Figure 2 provides another aggregation that examines the robustness of fit by considering all domains equally—a macro average of perplexity over domains: $|D|^{-1} \sum_{d \in D} \text{perplexity}(d)$ for domain set $D$. By contrast *ordinary perplexity* is essentially an exponentiated micro average over the domains implicitly selected for during corpus curation. Macro averaging lets all marked domains have equal say on the model's performance, instead. To make these macro averages more easily interpretable, we examine them separately per source.

The most striking pattern that emerges with per-source macro averages is the high, and sometimes non-monotonic, perplexity of the 3 baselines trained on only Common Crawl data (C4, MC4-EN, FALCON REFINEDWEB). This is particularly apparent for the C4 model evaluated on REDPAJAMA, where the macro average is dominated by perplexity up to 391,171 on the *arXiv* domain. Similar spikes occur for the FALCON REFINEDWEB and MC4-EN models, with perplexity of 21,652 and 1,409 respectively, on the Max music programming language domain in DOLMA-100-PROGRAMMING-LANGUAGES. These domains contain large amounts of non-natural language, in the form of LaTeX and other code data. These spikes stand out from the stable and monotonic improvement observed in the other 3 baseline models. While these Common Crawl baselines spike on different domains, it appears they are more susceptible to these extreme gaps in fit to *some* domains. Perhaps this occurs because of a lack of exposure to specific types of language completely filtered due to having only one set of cleaning filters applied to a single source of data.

---

[7]We do not decontaminate code as its paragraphs (lines) are short and often repeated.

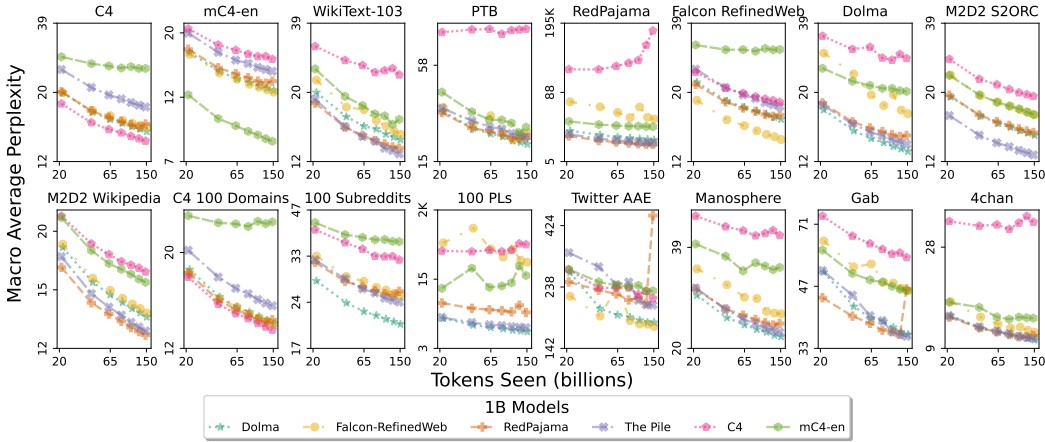

Figure 2: Perplexity macro averaged over any domains within each of the 16 top-level data sources (§2) in PALOMA, for each baseline model. Evaluating on one monolithic corpus, such as C4, does not tell the complete story of model fit. PALOMA lets us see when trends differ from one distribution of language to another. For instance, the 3 baselines trained on only Common Crawl data (C4, MC4-EN, FALCON REFINEDWEB) exhibit high perplexity, sometimes with non-monotonic scaling over tokens seen, on specific evaluation sources such as REDPAJAMA, and DOLMA-100-PROGRAMMING-LANGUAGES.

In contrast, the baselines that include curated non-webscraped text sources (DOLMA, THE PILE, and REDPAJAMA) have a relative gap in perplexity that is highly stable through the course of training. This would imply that short training runs on a subsample of such pretraining corpora may be predictive of the LM fit of specific sources after much longer training. To address one exception, the REDPAJAMA baseline often spikes on its final checkpoint, sometimes dramatically as in TWITTERAAE. A possible explanation is that this checkpoint falls very soon after the model's training loss recovers from a small spike.

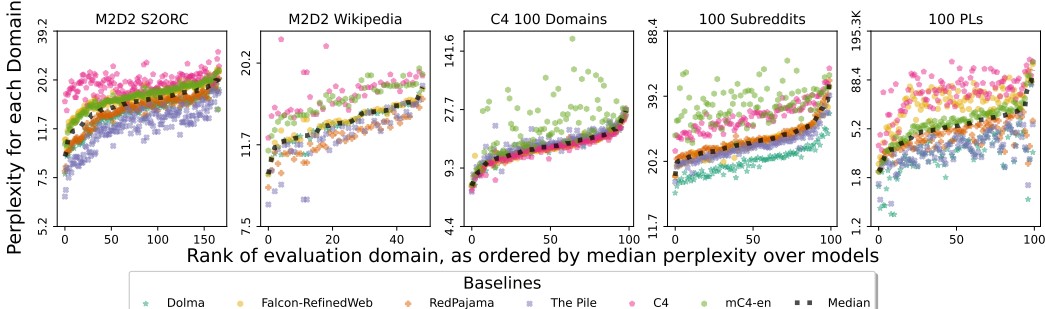

Figure 3: For each source with at least 10 domains, each point visualizes perplexity on a single domain for a fully trained model. Domains are ordered by median perplexity of that domain over all models. Gaps between some baselines are highly consistent across domains (e.g., REDPAJAMA and THE PILE baselines on DOLMA-100-SUBREDDITS). Other models (often pretrained on just Common Crawl data) exhibit noisy gaps that do not follow the trend in median domain difficulty (e.g., the MC4-EN baseline on C4-100-DOMAINS).

**Perplexity per domain ordered by median perplexity** We can visualize each perplexity separately for each domain to surface gaps in fine-grained LM fit. In Figure 3, we arrange the domains by their median perplexity over the baselines, as this order gives some sense of the intrinsic difficulty of a domain. We can then see which baselines follow this order, differing only by a consistent offset, and which have gaps that are more idiosyncratic to each domain. Again we see that when baselines have irregular gaps from the median these are most frequently baselines pretrained on only Common Crawl.

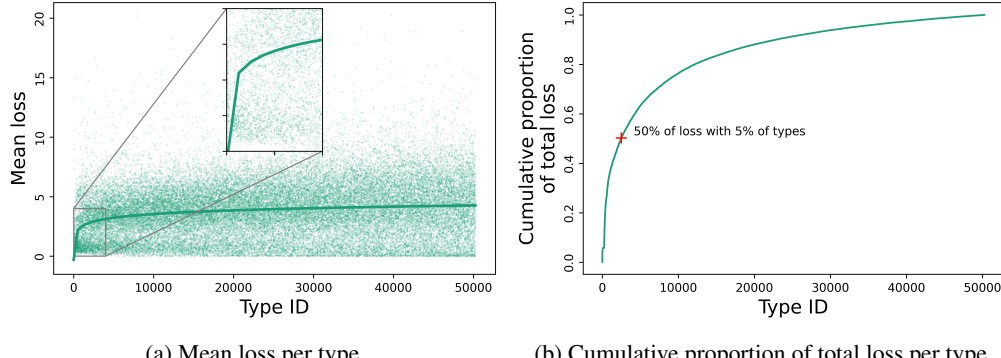

(a) Mean loss per type

(b) Cumulative proportion of total loss per type

Figure 4: Mean and total loss per vocabulary type, i.e., specific strings in the vocabulary. While high-frequency types (which have low IDs) tend to have a low *average* loss as shown by a log-linear regression (a), they contribute a substantial part of the *total* loss, simply by virtue of their frequent occurrence in the data (b). The figure shows the distributions for Pythia-7B [Biderman et al., 2023] on C4-100-DOMAINS, but the overall picture is consistent for different models and sources.

The notable exception is THE PILE baseline on M2D2 S2ORC and DOLMA-100-PROGRAMMING-LANGUAGES, which has erratic gaps substantially below the median, perhaps indicating that baseline is benefiting from exposure to specific domains and not others rather than only a overall facility for scientific papers and code. The erratic-gapped Common Crawl baselines, by contrast, are all worse than median perplexity, suggesting that they may have complete gaps in exposure to features of certain domains that are not recovered through generalization.

## 4.2   Common Vocabulary Types Dominate Perplexity

Here we present a single conclusion from a second case study; see Appendix D.2 for further analysis. So far we have examined perplexity aggregated over tokens. Another approach is to measure average likelihood per vocabulary *type*, i.e., the strings that are represented in the vocabulary of a model, in contrast to occurrences of these strings in some corpus, called *tokens*.[8]

**Few vocabulary types account for most of the loss measured in perplexity**   How much do specific *types* contribute to perplexity aggregated per token? To answer, we start by analyzing the total loss mass added by types, as a function of their IDs. Smaller IDs correspond to more frequent types in the GPTNeoX-20B tokenizer training data [Sennrich et al., 2016, Black et al., 2022], and we find an overall moderate to strong correlation between IDs and frequencies in the evaluation data of PALOMA as well (Pearson's $r$ averaged across domains: $-0.522\pm0.087$). Crucially, frequency has a strong impact on the total loss mass associated with individual types: while the *average* loss is lower for the high-frequency types (Figure 4a), the *total* loss is higher, resulting in a situation where 5% of the types already cover roughly 50% of the overall perplexity (Figure 4b). Thus, perplexity is strongly influenced by a relatively small set of high-frequency types. This finding provides further evidence that reporting only aggregated perplexity values neglects more subtle dynamics visible through fine-grained analysis (i.e., sources, domains, vocabulary types) in PALOMA.

## 5   Conclusion

We believe that evaluations of language modeling fit provide an important view of performance that has been neglected in recent LM research and development. Perplexity cannot be naïvely applied to language modeling at this scale due to challenges such as benchmark contamination. However, these obstacles are worth overcoming as perplexity offers several advantages not afforded by downstream evaluations. Instead of constructing tasks from scratch, we can rely on the ecological validity of

---

[8]See Appendix B for a more formal definition of average likelihood per vocabulary type

real-world data drawn from known sources. Finding the best ways to evaluate model fit to a collection of documents creates an interface for other fields to contribute to the evaluation of language models. Without needing to understand LM architectures, researchers in other fields can collect corpora representing domains of interest that LM researchers would not know to consider. Once such sources are identified, evaluations can be updated over time by simply scraping more data, unlike downstream tasks where expensive annotation would be required.

Further, we hope that PALOMA provides controlled results for study of when perplexity evaluations are or are not predictive of downstream performance [Liu et al., 2022, Tay et al., 2021, Ganguli et al., 2022, Xia et al., 2022, Gadre et al., 2024, Du et al., 2024]. In Appendix A, our preliminary investigation reveals that different PALOMA sources are correlated with some downstream tasks and anticorrelated with others. This contrasts with the assumption in much scaling literature that lower perplexity always indicates better downstream performance. While we do observe that LM loss reduces with scale across most domains (Appendix D.1), the fit of this relationship and the relationship of loss to downstream performance will both differ for each pretraining and validation distribution as observed by Gadre et al. [2024]. This means that one cannot simply find which fine-grained perplexity domains correlate with one's favorite task and then hillclimb on those. Instead further investigation with pretraining experiments across a wide range of scales and data recipes is needed to understand when reductions in perplexity are being driven by superficial overlaps of train and validation distributions or by learning features relevant to downstream use.

## 6  Limitations and Future Work

The largest limitation of PALOMA is that we elect to focus just on the language modeling of English and code data. We select this scope as most current LMs also focus on theses types of data. However, we strongly encourage future work to explore how language model fit to fine-grained domains behaves within and across other languages.

Proper use of perplexity as a metric must take into account its limitations. We believe perplexity is best used to show what a model *is* learning rather than what it *should* be learning. For instance we find that perplexity on the 3 fringe datasets are tightly related to average document lengths, with the short tweet-like posts in GAB CORPUS receiving high perplexities while the long concatenated threads of posts in 4CHAN CORPUS and MANOSPHERE CORPUS provide greater context and lower perplexity. At this level of aggregation, differences in surprise between these domains likely have little to do with model fit to specific types of toxicity and more to do with how models use extremely short or long contexts. In our case study in §4.2, we demonstrate that often it is more appropriate to decompose measures of surprise over specific strings within a corpus, rather than aggregating over all text in a domain. We hope that by surfacing the average likelihoods of specific strings in the vocabulary, PALOMA can enable future work on metrics that better measure the fit of models to the features of language in specific domains that humans find most salient.

We also highlight guidelines for evaluating with perplexity (§3). In particular we believe decontamination of benchmark leakage and balancing variance induced by subsampling across domains are both challenging concerns requiring further investigation. For each of these we have proposed one simple and scalable mitigation (see Appendix C.1.1 and C.2.1 for further details), but future work should explore alternatives and measure their efficacy.

PALOMA curates and standardizes the text datasets with the most fine-grained domains readily available from existing metadata. As such, our definition of domains by metadata is necessarily heuristic. Some overlapping domains in PALOMA appear in multiple sources, such as academic papers. Though DOLMA and REDPAJAMA process academic papers differently, the subcorpora on academic papers in each source represent different approximations of the same or very similar domains. However for the sake of simplicity, we make the reductive assumption of counting all 546 domains in PALOMA as fully distinct. We hope that future work will explore novel means of identifying fine-grained domains and separating distribution shifts in language due to differing authorship or differing data collection processes.

## Acknowledgements

We thank Nishant Subramani, Akhila Yerukola, Rodney Kinney, and Ari Holtzman for fruitful conversations. The experimental components of this work were made possible through a partnership with AMD and CSC, enabling use of the LUMI supercomputer.

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

# Appendices

## A  Downstream Correlation Analysis

In Table 3 we provide the Spearman's rank correlation between the ranking of our 6 baseline models' final checkpoints by each of the 16 PALOMA sources and by each of the 8 downstream evaluations used in OLMo [Groeneveld et al., 2024]. These are Arc (both Easy and Challenge) [Clark et al., 2018], Boolq [Clark et al., 2019], Hellaswag [Zellers et al., 2019], Openbookqa [Mihaylov et al., 2018], Piqa [Bisk et al., 2020], Sciq [Welbl et al., 2017], and Winogrande [Sakaguchi et al., 2021].

These results provide some indication of relationships between specific perplexity sources and downstream tasks, such as m2d2 wikipedia and Arc Challenge or Hellaswag and c4-en. Most importantly, it is apparent that no single perplexity evaluation correlates well with all downstream tasks, suggesting the importance of evaluating across a range of diverse perplexity evaluations rather than a single monolithic validation loss.

However, we caution against reading too far into these correlations without further pretraining experiments across a greater range of compute scales and data mixes. For our set of 6 pretraining experiments, the correlation of rankings by the *same* downstream tasks between adjacent model checkpoints is only a moderate 0.513 when averaged over tasks and checkpoint pairs. This suggests that the differences in downstream performance between these mixes at this scale are not stably discernible on their own regardless of correlation to perplexity. We hope that users of our benchmark will create controlled pretraining experiments at larger scales with more distinct data mixes whose downstream rankings are more consistently discernible.

| | Arc Challenge | Arc Easy | Boolq | Hellaswag | Openbookqa | Piqa | Sciq | Winogrande |
|---|---|---|---|---|---|---|---|---|
| c4-en | 0.38 | 0.26 | **0.60** | **-0.77** | -0.49 | -0.41 | -0.23 | -0.03 |
| mc4-en | -0.20 | -0.49 | 0.09 | 0.03 | **0.83** | -0.23 | 0.41 | **0.61** |
| wikitext | **-0.67** | -0.37 | 0.09 | **0.83** | 0.03 | **0.58** | -0.41 | **-0.61** |
| ptb | -0.49 | **-0.71** | 0.43 | 0.49 | 0.37 | 0.12 | -0.12 | -0.26 |
| redpajama | **-0.52** | -0.43 | -0.20 | **0.94** | 0.20 | 0.49 | -0.17 | -0.46 |
| falcon-rw | **-0.55** | -0.31 | **0.94** | -0.14 | -0.03 | 0.12 | -0.46 | -0.26 |
| dolma | -0.38 | -0.49 | 0.31 | **0.54** | 0.26 | 0.06 | -0.35 | -0.20 |
| m2d2 s2orc | -0.46 | -0.31 | 0.14 | **0.71** | 0.09 | 0.29 | -0.49 | -0.38 |
| m2d2 wikipedia | **-0.78** | **-0.60** | 0.20 | **0.77** | 0.14 | **0.64** | -0.17 | **-0.67** |
| c4 100 domains | 0.23 | 0.37 | **0.66** | **-0.83** | **-0.60** | -0.23 | -0.32 | -0.12 |
| 100 subreddits | -0.23 | -0.14 | **0.54** | 0.20 | -0.09 | -0.06 | **-0.67** | -0.20 |
| 100 PLs | -0.23 | **-0.54** | 0.03 | **0.66** | 0.43 | -0.03 | -0.12 | -0.06 |
| twitterAAE | 0.06 | 0.31 | **0.60** | -0.37 | -0.20 | -0.35 | **-0.72** | 0.23 |
| 4chan | -0.38 | -0.49 | 0.31 | **0.54** | 0.26 | 0.06 | -0.35 | -0.20 |
| manosphere | -0.38 | -0.49 | 0.31 | **0.54** | 0.26 | 0.06 | -0.35 | -0.20 |
| gab | -0.20 | -0.03 | 0.09 | 0.49 | 0.14 | -0.03 | **-0.61** | 0.06 |

Table 3: Spearman's rank correlation of our 6 baseline models between PALOMA perplexity evaluations and downstream tasks. Values greater than abs(0.5) are bolded for emphasis.

# B   Additional Metrics

This section details two additional metrics that can be used in PALOMA.

**Bits per byte**   When comparing results where model vocabularies must differ, for instance research to improve tokenizers, PALOMA follows Gao et al. [2020] in using bits per byte (BPB). This metric normalizes the log likelihood $\ell$ over documents by the count of UTF-8 encoded bytes in the corpus, $B$:

$$\text{BPB} = \frac{1}{B}\log_2(e^{-\ell}) = \frac{-\ell}{B\ln(2)}$$

**Average likelihood per vocabulary type**   Both perplexity and BPB can be driven by strings that occur frequently, dominating subtler differences in performance on other strings. An alternative is to measure surprise over all occurrences of specific strings instead. A set of strings particularly important to the model's functioning are the strings represented in the model's vocabulary. Following conventional NLP terminology, we call the elements of the vocabulary *types* in contrast to occurrences of these strings in some corpus, which are called *tokens*. When running inference in PALOMA we record $\mu(\ell_v)$, average likelihoods over the whole corpus for each type $v$, as well as $\mathbf{T}_v(N)$, the count of occurrences of that type over the whole corpus (with indicator function $\mathbb{1}(\cdot)$):

$$\mu(\ell_v) = \frac{1}{\mathbf{T}_v(N)}\sum_{t \in N}\sum_{i}^{|t|}\mathbb{1}(v = t_i)\ln p(t_i|t_{<i})$$

# C   Experimental Controls

Here we discuss the details of the experimental controls (introduced in §3) that we implement to meet our guidelines for rigorous perplexity evaluations. We distinguish controls that must be applied during model training and controls that are applied at inference time.

## C.1   Training Controls

### C.1.1   Decontamination

A basic tenet of machine learning is that for evaluation to accurately represent performance, training and test data need to be non-overlapping. However, large pretraining corpora are known to contain evaluation data and large models are known to memorize training data [Dodge et al., 2021, Elazar

| Dataset | Document Removal Rate |
|---|---|
| DOLMA | 0.062% |
| REDPAJAMA | 0.099% |
| THE PILE | 2.753% |
| FALCON REFINEDWEB | 0.733% |
| C4 | 0.010% |
| MC4-EN | 0.002% |

Table 4: Decontamination removal statistics for the corpora with which we train our 6 baseline models. We remove any training document with any paragraph marked as contaminated against PALOMA.

et al., 2023, Carlini et al., 2022]. Lee et al. [2022] show in their second figure that models under-estimate perplexity on evaluation documents with near duplicates in the training corpus by several points relative to models with those duplicate training documents removed. Thus benchmarks of language modeling should actively remove contaminated training data, rather than just partitioning held out splits by documents, assuming no documents overlap. THE PILE applies document-level deduplication to two of their 22 domains before splitting held-out data, but its designers note that this does not prevent leakage of evaluation data more generally [Gao et al., 2020]. Furthermore, spans of contaminated text within larger unrelated documents can still contribute to overestimation of performance, so decontamination should be conducted at a sub-document level. To our knowledge, PALOMA is the first language modeling benchmark to require removing training data that is contaminated with respect to evaluation data.

To mitigate contamination of our benchmark, we develop an approach for removing contamination from training data at the scale of pretraining corpora of trillions of tokens. We use a Bloom filter [Bloom, 1970] as implemented in Soldaini et al. [2024] to match training text that is contaminated with respect to the evaluation data. We employ this approach rather than the minHash or suffix array approaches used by Lee et al. [2022] and other deduplication work, as our approach is much more lightweight: the minHash approach would require pairwise computations, $O(|X_t||X_e|)$ between all training texts, $X_t$, and evaluation texts, $X_e$, where our approach runs a constant number of hashes, $K << |\mathcal{X}_e|$, over all texts in $O\left(K(|X_t| + |X_e|)\right)$. Meanwhile the implementation of the suffix array approach of Lee et al. [2022] requires memory usage proportional to the size of the pretraining corpora. Since we aim to encourage researchers using our benchmark to run this decontamination on their pretraining data, we opt to minimize cost and engineering complexity.

Using our approach to find text matches, we mark contamination in the following way. We match text at the paragraph level, i.e., newline separated spans of text. This granularity strikes a balance between, on one hand, examining only full documents, which can miss contamination embedded in novel documents, and, on the other hand, all n-grams of a given size, where the size of the n-grams must be carefully set. Instead paragraph matching leverages this naturally occurring unit of language, although this heuristic has its own limitations especially in domains such as code or poetry, where line separation is handled very differently from prose. To avoid coincidental collisions in the space of small strings, we ignore matches in paragraphs smaller than 13 unicode segmented tokens [Unicode, 2023], as 13 is the n-gram sized used in contamination checks in Brown et al. [2020] and Rae et al. [2021]. Similarly, we ignore paragraphs composed of only punctuation, spaces, and emoji, as, unlike words, these can be arbitrarily repeated when used as formatting, leading to high frequency n-grams greater than our 13-gram threshold. Lastly, as code data consists almost entirely of short and often repeated lines, we forgo any decontamination on these sources (DOLMA-100-PROGRAMMING-LANGUAGES and the THE STACK domain of DOLMA). We leave the question of how to properly decontaminate code data to future work.

Having marked contaminated paragraphs, we now take the conservative measure of removing whole documents if they contain *any* contaminated paragraph. This has the added benefit of not disrupting the continuity of text within documents, which excising paragraphs would do. Applying this approach to the datasets on which we train 6 baseline models results in the removal rates shown in Table 4. While these vary by orders of magnitude from dataset to dataset (with THE PILE perhaps receiving a

higher removal rate due to the intentional oversampling in that dataset), this approach removes at most 2.753% of documents, making it feasible to apply without dramatically reducing training dataset size. Nevertheless, care should be taken to examine removal rates when applying this approach to new datasets.

Another limitation arises from our use of documents as a fundamental unit of data. This impacts our decontamination approach, since we remove whole documents that have any paragraph marked as contaminated to avoid mangling documents by excising individual paragraphs. Such an approach tends to disproportionately remove long documents that are frequently quoted, which may include seminal works (e.g., Martin Luther King's "I Have a Dream" speech) that actually deployed models should be familiar with. The purpose of PALOMA, however, is to enable controlled research on the science of language modeling, but production models should likely use caution in applying this decontamination technique.

### C.1.2   Data Order

Another decision that affects language modeling experiments is the order of training documents. While intentionally designing curricula by ordering training data to improve performance is an area of active research (Bengio et al., 2009, *inter alia*), most LMs simply randomize the training order. In this case greater comparability between experiments with the same dataset can be achieved if the same random order is used for all models. This also facilitates research that examines exactly what data a given model checkpoint has seen or not seen at that point in training. No previous language modeling benchmarks require the fixing of training order.

As contemporary LMs train on instances that are themselves concatenations of training documents up to the maximum sequence length of the model, to fix the order of training data one cannot simply fix the order of documents but must train on the same concatenated instances. Achieving this requires not just a fixed random seed for training instance shuffling, but also adopting the same tokenization and maximum sequence length. Further fixing the number of instances in each gradient update would be required for fully identical training, however this is onerous for experiments that may be run on different hardware requiring different batch sizes. A compromise instead is to ensure that training code feeds instances into gradient steps in a deterministic shuffled order, so the relative ordering of data remains the same even if a given instance may fall in different gradient updates. In conclusion, we adopt the most direct way of controlling data order—we recommend using the same training code that we use to pretrain our baseline models.

### C.2   Evaluation Controls

### C.2.1   Subsampling

There is no shortage of text that can be used to estimate perplexity, so we must choose how much to evaluate based on a tradeoff of inference cost and metric stability over different subsamples. The value we ultimately care to estimate is the perplexity of the model on all the available data, not just a subsample. Much existing work considers the estimation of other information theoretic quantities such as entropy and mutual information (Paninski, 2003 *inter alia*), so the estimation of perplexity should likewise be treated with care, for instance in subsampling evaluation data. Previous benchmarks subsample uniformly over the whole corpus, leaving some domains represented by very little data. M2D2 mitigates this by an ad hoc minimum size, but this still leads to domains with different sizes. PALOMA takes a first step towards controlling for subsampling induced variance in perplexity estimation by using a stratified subsample across domains and providing a preliminary empirical measure of metric bias and variance extrapolated from one domain.

In Figure 5, we evaluate perplexity on data from C4 using Pythia 1.4B [Biderman et al., 2023] while varying the size of the evaluation subsample and training checkpoint. Each point in this figure represents the mean of perplexity on 20 different uniform subsamples and standard deviation is represented by the shaded region. As we expect, for a given checkpoint standard deviation shrinks as the evaluation subsample gets larger. More subtly, standard deviation shrinks as the model is trained

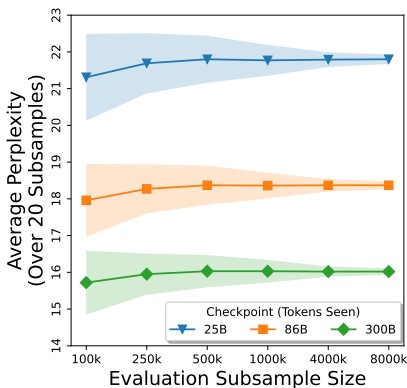

Figure 5: Average perplexity and standard deviation over 20 subsamples of C4 validation data using Pythia 1.4B checkpoints. We find that variance in perplexity over subsamples of evaluation data decreases steadily as evaluation samples grow.

on more data. This second observation matters if we want to measure model performance throughout training. Lastly note that the mean value is relatively stable over different evaluation subsample sizes, though a slight downward trend appears at the smallest subsample sizes.

The stable trend of subsample size and variance in perplexity allows us to estimate how much perplexity numbers might change if a different subsample of the same size were drawn. Furthermore, when preparing splits for perplexity evaluation across many domains, it would be best to size for a similar level of metric variance. Most often perplexity evaluation data is subsampled uniformly over the original distribution of domains in a source, resulting in more or less tokens from each domain in the evaluation data based on how well represented they are in the corpus. We instead employ stratified sampling, in which all sources with marked domains are partitioned by domain and a uniform sample of the same size is taken from each partition. Specifically, documents are sampled from each domain until the same target number of tokens is reached. This helps ensure that no domains are lost or very small after subsampling.

As a small first step towards more principled subsampling, we set the target subsample size based on the simplifying assumption that our metric variance results on C4 hold for other domains and models. Extrapolating our observations, we aim to subsample each split to a minimum of 1 million tokens per source and a minimum of 100 thousand tokens per domain. All datasets with domains are subsampled to 100 thousand tokens per domain other than MANOSPHERE CORPUS which we treat as a single-domain source, ICE which was included in early versions of Paloma in entirety for comparability to its use in HELM, and DOLMA which we subsample at a higher target of 500 thousand tokens per domain. A few sources fall below our thresholds, with WIKITEXT-103, PENN TREEBANK, and TWITTERAAE being smaller than 1 million tokens per split despite being included in their entirety, and REDPAJAMA having only 7 domains leading to 700 thousand tokens per split. We show the final token statistics in Table 1.

If extrapolation from the trends we observed holds, perplexities on sources will be drawn from a distribution over subsamples with less than 1 standard deviation even at very early stages of training. Meanwhile, results on domains will be drawn for a similarly stable distribution by the end of training. This is admittedly a heuristic simplification, as the relationship between variability and subsampling will also likely depend on other factors such as average document length and heterogeneity of the source data, as well as the power of the model being evaluated. We must leave it to future benchmarks to explore these questions as the requirement of decontaminating pretraining data against evaluation data means any change to the evaluation data necessitates costly rerunning of pretraining of all baselines.

Another limitation arises from our use of documents as a fundamental unit of data. When subsampling although we balance the number of tokens used to represent each domain, we still sample documents

until that target token count is reached. Concretely, this means that some domains, especially books, are represented by only dozens of documents, which likely does not capture the full distribution of the domain as well as many smaller documents might.

### C.2.2 Vocabulary

Perplexity per token is not comparable between models with different vocabularies [Jelinek, 1998] or, by extension, different tokenizers [Mielke, 2019]. Since models distribute probability over a vocabulary of tokens, models with larger vocabularies will tend to have higher perplexities than ones with smaller vocabularies. Where possible, the most rigorous solution is to impose one vocabulary on all experiments, allowing perplexity to be directly compared. Some lines of research, such as improving tokenizers, require comparisons of LM fit *across* vocabularies. This is possible by normalizing likelihood by a segmentation intrinsic to the text such as characters or bytes [Mielke, 2019]. THE PILE [Gao et al., 2020] proposes BPB (Appendix B) as the best compromise when tokenizers are not identical, an approach we adopt as well. PALOMA further establishes a standard tokenizer and vocabulary for experiments that do not need to change this experimental variable.

Where possible we control by the simplest approach of using the same vocabulary: the vocabulary used in GPT-NeoX-20B [Black et al., 2022] with 3 special tokens added by DOLMA for masking personally identifiable information. Note that when vocabulary is fixed this is essentially a training control, as the model must be pretrained with this vocabulary. Nevertheless we mark this as an evaluation control, as we provide an option applied at inference time for making comparisons of models already pretrained with different vocabularies. Specifically, we follow THE PILE [Gao et al., 2020] and use BPB. In theory BPB may still present issues in comparability as it only includes likelihoods of the specific sequences produced by a given tokenizer, e.g., *rain ##ing* for the text *raining*, and not the marginal probability over all valid sequences in that vocabulary which would produce the identical text, e.g., *ra ##in ##ing* and so on (Mielke, 2019, Cao and Rimell, 2021; see also Hofmann et al., 2021). Models with a larger event space of possible sequences representing the same text will be at a disadvantage if they assign any non-zero probability to these valid predictions ignored by the metric. However, it has been shown empirically that the difference between the marginal probability over all valid sequences and the likelihood of the sequence produced by the tokenizer is small [Mielke and Eisner, 2018] and typically lower than 0.5% [Chirkova et al., 2023]. So in conclusion, we encourage those using PALOMA to opt in to our fixed vocabulary, or make comparisons involving models with different vocabularies in BPB.

### C.2.3 Evaluation Format

While perplexity is clearly defined as a function of the likelihood assigned by a model to a set of sequences, the manner in which that likelihood is computed may vary depending on how inputs are formatted for the model. THE PILE [Gao et al., 2020] identify one possible variation: inferring test documents as separate inputs or concatenating them together to fill a single input. Meanwhile, Press et al. [2021] point out that documents larger than the maximum sequence length can be split either with or without overlap.

We follow the input format established by THE PILE [Gao et al., 2020]. In this format, documents are evaluated individually, e.g., "*<BOS>document 1*" then "*<BOS>document* 2", rather than packed into concatenated maximum sequence length inputs, e.g., "*<BOS>document 1<BOS>document 2<BOS>...*", where *<BOS>* is a special token for demarcating sequences. The latter concatenated approach is still often used as it takes the same preprocessing as is most commonly used for training data and is thus convenient for measuring validation loss during training. However, in Appendix H we find preliminary evidence that the predictability of variance from subsampling observed in Appendix C.2.1 breaks down for concatenated inputs. We also believe that evaluating documents individually more closely mirrors how models are used in practice at inference time. Providing more than one document at a time through concatenation is essentially a form of few shot in context learning for language modeling, as it allows the model to condition on information shared between

concatenated documents when they are all drawn from the same domain. This is perhaps an interesting task formulation of its own but one that should be undertaken intentionally.

Moreover, following THE PILE, we split documents longer than maximum sequence length into disjoint inputs. This is also described by Press et al. [2021] as *nonoverlapping inference*. It is contrasted with *sliding window inference* in which some amount of overlapping tokens are included as context in maximum-sequence-length windows to prevent an unrealistic lack of conditioning for tokens in the middle of a document appearing shortly after a multiple of the maximum sequence length. However, a sliding window requires re-encoding overlapping tokens, making nonoverlapping inference the most efficient approach to computing perplexity.

## D    Additional Case Studies

In this section, we present additional case studies to explore analyses possible with PALOMA. Previously in §4.1, we use our 6 baseline 1B models that vary only in which common corpus they are pretrained on to isolate the effect of data composition on LM fit. In §4.2 we introduced the observation that most loss occurs on the most common vocabulary types, which we now expand on in Appendix D.2 by analyzing performance dynamics of different vocabulary types. First, in Appendix D.1, we examine how scaling dynamics differ over the breadth of domains in PALOMA.

Results in the appendix include two additional sources THE PILE [Gao et al., 2020] and ICE [Greenbaum and Nelson, 1996], however access restrictions on these datasets prevent us from rehosting them. As such we have removed them from the body of our paper, but still share our findings on these datasets as auxiliary results not part of PALOMA.

### D.1    Scaling Improves Domain Fit Unequally

We return to the question, does rising performance lift all domains? That is, does the sign of scaling trends observed in previous work [Kaplan et al., 2020, Hoffmann et al., 2022] hold across all domains? And if so, do some domains still capture most of the improvement while others stagnate?

#### D.1.1    Scaling Tokens Seen

In Figure 6, we study the impact of increased training on domain fit. We make use of the finding that the logarithms of loss and tokens seen trend linearly Kaplan et al. [2020], and make an estimate of improvement based on the slope between two empirical observations of perplexity (*ppl*), with some initial and final number of tokens, $i$ and $f$, seen by checkpoints of a model $\theta$:

$$\Delta_t(i, f) = \frac{\ln(\ln(\mathrm{ppl}(\theta_i))) - \ln(\ln(\mathrm{ppl}(\theta_f)))}{\log_{10}(f) - \log_{10}(i)}$$

Specifically, we plot $\Delta_t(\sim 20B, \sim 150B)$ for each domain in ascending order for each of our 6 baselines.[9]

**On some corpora, more pretraining worsens fit on some domains**    Baselines trained on C4 and MC4-EN worsen with longer training on 65 and 43 domains respectively. Other than these two baselines, only 6 other pairs of models and domains see such a deterioration. Among these 6 pairs only the REDPAJAMA baseline exceeds $\Delta_t(\sim 20B, \sim 150B) > 0.1$, likely due to the previously noted spike in training loss near the final checkpoint of this model. It is unclear why the other baseline trained on only Common Crawl data, FALCON REFINEDWEB, does not also exhibit erratic behavior this time, though possibly its cleaning heuristics avoid removing content important to these domains that the other two models' cleaning heuristics do remove.

---

[9]Note that the precise number of tokens seen by a given checkpoint does vary slightly between baselines, as these were run on heterogeneous hardware requiring slight differences in batch size.

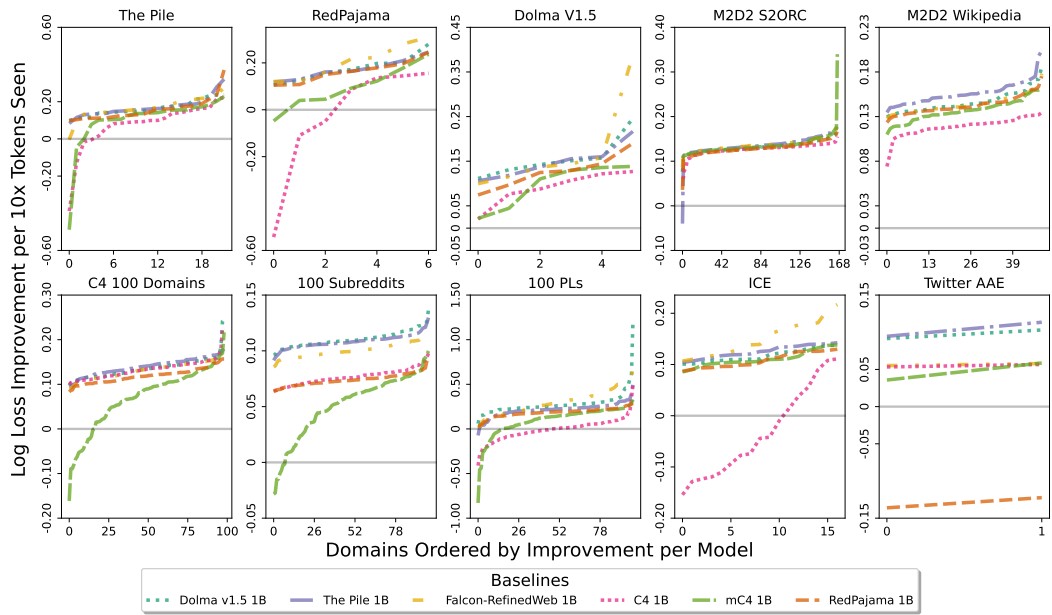

Figure 6: As log loss and log tokens trend linearly, we estimate reduction in log loss per $10\times$ increase in tokens seen based on the slope between $\sim$20B and $\sim$150B checkpoints. We report this rate of improvement for each domain in ascending order per baseline model. This reveals that for some models and domains, loss actually increases with further training. However, excepting just 6 model-domain pairs, all baselines other than C4 and MC4-EN improve on all domains with a similar range between most and least improvement. Even among these, the median difference in improvement between most and least improved domains has nearly twice as fast improvement for most improved domain.

**Even for corpora where fit consistently improves, the rate of improvement is unequal**   On the vast majority of domains, fit does improve with increased training. However rates of improvement, $\Delta_t(\sim 20B, \sim 150B)$, range substantially. Examining the median difference in improvement between most and least improved domains shows 1.57x improvement for most improved domain, and this gap grows to 1.94x when excluding the C4 and MC4-EN baselines.

**Slow improvement on a domain is not always unwanted, but surfaces dynamics of model learning**   Having identified the most and least improved domains, we visualize perplexity curves of 3 examples each demonstrating a different interpretation in Figure 7. On the left plot we see that sometimes fit can actually worsen on one domain while improving on another domain, in this case perhaps due to content filters in MC4-EN pretraining data blocking terms frequently used in discussion about dating and sexuality. But even when fit improves on both domains as in the middle plot, the rate of improvement can be slower for one than the other, possibly reflecting differences in the quantity or heterogeneity of earth sciences or visual arts content in DOLMA. However, the right plot shows that the least improved domain can actually outperform the most improved domains in terms of absolute perplexity, in this case perhaps representing saturation of performance on the DM Mathematics domain. Further examples are provided in the Appendix in Figure 13. Ultimately, our goal is not to frame unequal improvement as a problem that needs to be fixed, but rather it is way to surface subtler dynamics in language model learning.

### D.1.2   Scaling Model Parameters

While the 6 baseline models that we pretrain ourselves are all 1B parameter models, we can use models of varying sizes from the Pythia model suite [Biderman et al., 2023] to examine the impact

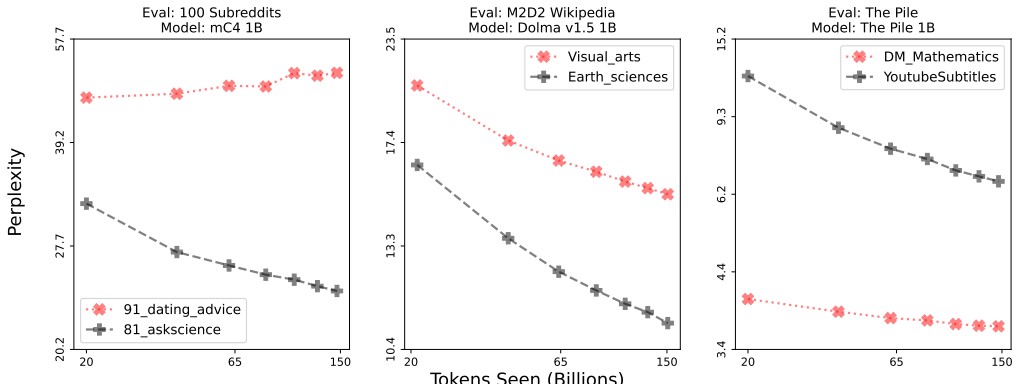

Figure 7: We examine 3 types of examples of most (black dashed) and least (red dotted) improved domains for 3 pairs of sources and models, where improvement is measured in terms of log loss per $10\times$ increase in tokens seen (see Figure 6). As on the left, fit to a least improved domain can actually worsen in absolute terms or, as in the middle, simply improve more slowly. On the right, we see that least improved domains may even be better fit in absolute terms. Unequal improvement between domains is not undesirable *a priori* but merits finer-grained examination, enabled by PALOMA.

of scaling model parameters on domain fit. As we note in §G, these models are not controlled for contamination but they do address all of our other guidelines.

**Increased parameter count sees consistently lower perplexity**    In Figure 8, we show the macro average of perplexity over any domains in each source (as we did in Figure 2) for 3 sizes of Pythia model. Not only does this always show an increase in performance with greater parameter count, but the relative differences between the performance curves are remarkably stable across all sources. Additionally, macro average perplexity decreases faster over number of tokens seen for larger models in all sources.

**Improvements from model size improve unequally for different domains**    In Figure 9 we perform the same analysis of improvement in log loss as before but this time with respect to log increase in non-embedding parameters, $\Delta_p(i, f)$. Specifically we plot $\Delta_p(85M, 805M)$ and $\Delta_p(805M, 6.4B)$ for the non-embedding parameter counts corresponding to the 160M, 1B, and 7B model sizes for each domain in ascending order per pair of models compared. This time scaling does universally result in improvements. However, the rate of improvement varies greatly from domain to domain. Examining the median difference in improvement between most and least improved domains shows $2.02\times$ improvement for the most improved domain, a similar gap to that seen on increases in tokens seen. Again, we stress that unequal improvement is not necessarily problematic, but rather it helps identify outlier domains that follow different scaling trends than the majority of the data. We offer examples of most and least improved domains with respect to increase in model size in the Appendix in Figure 14.

Taken together, the results presented in this case study demonstrate the need to decompose evaluations of LM fit along domains. They show that it is not the case that models improve at uniform rates across domains for a given increase in scale. We leave it to further work to examine when these inequalities are or are not desirable and what interventions can help prevent stagnation of LM fit to certain domains.

### D.2    Common Vocabulary Types Dominate Perplexity, Others Have Inverse Scaling

Previously in §4.2 we noted that few vocabulary types account for most of the loss measured in perplexity. Now we continue to explore the dynamics of average likelihood per vocabulary *type*, i.e., the strings that are represented in the vocabulary of a model (Appendix B).

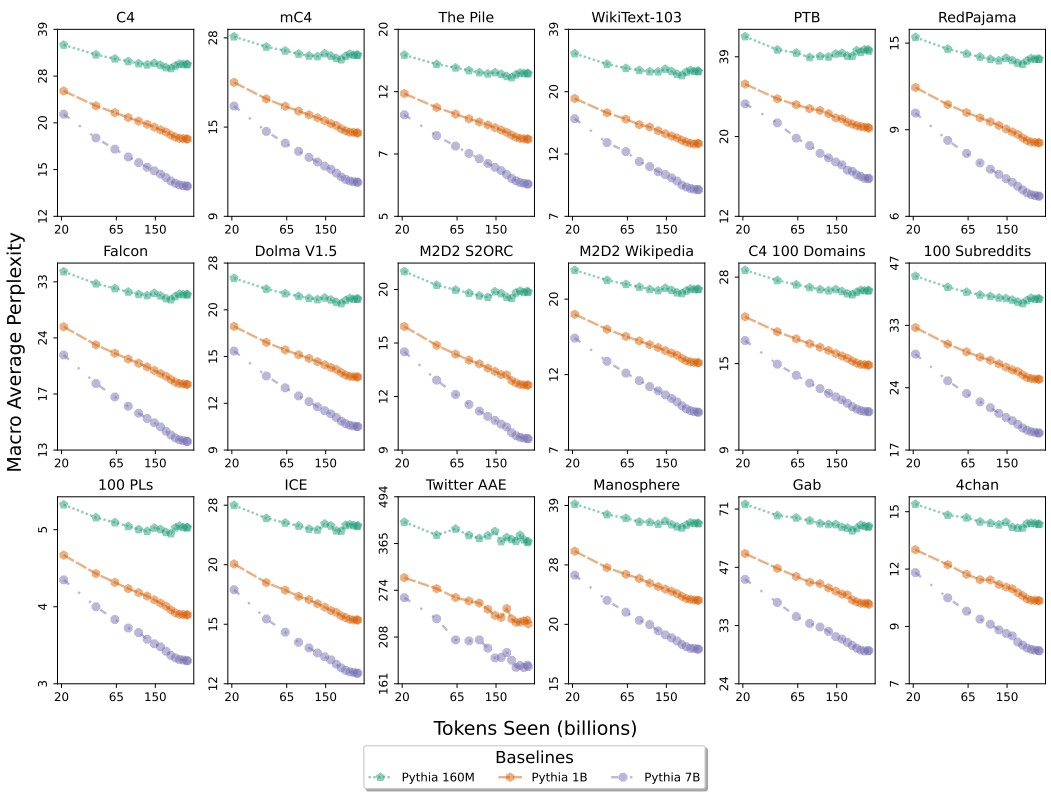

Figure 8: Perplexity macro averaged by domain in each source for checkpoints of 3 Pythia model sizes. Note that these public models are not trained on decontaminated data, so these results should be treated with greater skepticism than the results on the 6 baselines that we train under experimental controls. Consistently across these sources, increases in number of model parameters improves perplexity and the rate at which perplexity improves per token seen.

**Some types are more surprising on average to larger models than smaller ones** Is there variation between models in terms of how much types contribute to perplexity? Put differently, if model $A$ has a lower aggregated perplexity than model $B$, can we conclude that it has a lower loss for all types? Conducting an exploratory analysis of Pythia-1B vs. Pythia-7B, we find that this is *not* the case: while Pythia-7B has a lower perplexity on all domains, there are always types that are better predicted by Pythia-1B (see Figure 10), with the average proportion of such types varying between 8.5% (C4-100-DOMAINS) and 32.1% (TWITTERAAE). As shown in Figure 11, the proportion of types on which Pythia-1B is better increases with ID, for all examined sources. In other words, while Pythia-7B is almost always better on high-frequency types, Pythia-1B is better on a substantial portion of low-frequency types. This pattern is not captured well by perplexity, which is influenced very little by the performance on such low-frequency types (see above). However, note that even in the high-frequency regime around 10% of types are better predicted by the smaller model. Many of those types also have a high frequency in the sources, indicating that our finding cannot be explained merely as a result of noisy measurements. For example, the pronoun *I* occurs 14703 times in ICE but its measured mean loss on the final checkpoint is lower for Pythia-1B than Pythia-7B.

**Lower average loss per type can be the result of several different training dynamics.** What does it mean specifically if Pythia-1B has a lower average loss on a specific type than Pythia-7B? Figure 12 shows, for each of the 16 sources, the training dynamics of an example type for which Pythia-1B is better than Pythia-7B after convergence. As can be seen, there are various patterns: sometimes there is a constant gap between the two models, with Pythia-1B being better from the very beginning (e.g., *Boat* in FALCON REFINEDWEB); sometimes Pythia-1B has a constant loss while

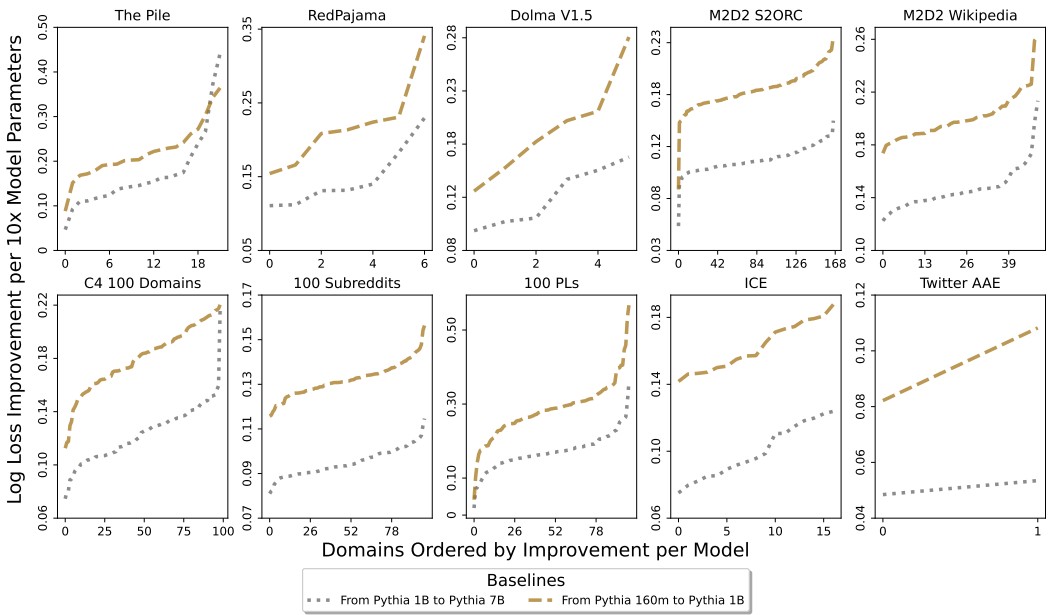

Figure 9: We estimate log loss improvement per $10\times$ increase in non-embeddings parameters based on improvement from Pythia-160M to Pythia-1B and from Pythia-1B to Pythia-7B on their final checkpoints. We report this rate of improvement for each domain in ascending order per compared model pair. These increases in model size always improve performance on each domain, but the median difference in improvement from least to most sees twice as fast reduction of loss.

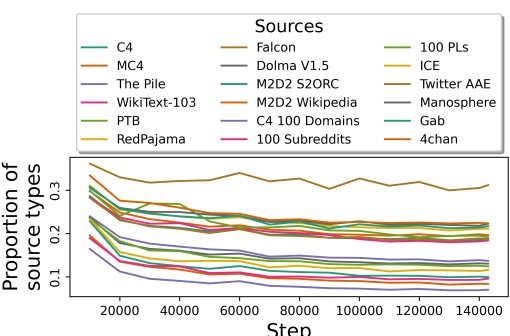

Figure 10: Proportion of types in each source for which Pythia-1B makes better predictions than Pythia-7B, as a function of training duration. The figure shows that for all examined sources, and even on the final checkpoint, a non-negligible proportion of vocabulary types is better predicted by the smaller model (i.e., Pythia-1B). This observation is particularly true for TWITTERAAE, where the proportion of such types is on average larger than 30%.

Pythia-7B is getting worse over time (e.g., *schedule* in DOLMA); sometimes Pythia-7B has a constant loss while Pythia-1B is getting better over time (e.g., *exchanged* in THE PILE); finally, sometimes Pythia-1B is decreasing its loss while Pythia-7B is increasing its loss over time (e.g., *BR* in C4). Especially the last pattern bears a resemblance with *inverse scaling* effects that characterize other aspects of LM behavior, where the performance gets worse rather than better with larger models [Mckenzie et al., 2023]. We are not aware of prior work describing the kind of type-level inverse scaling that we observe in this analysis.

**Some domains have more inverse scaling types than others** We also notice that there is further variation on the domains within the sources: for example, in TWITTERAAE (the source where the proportion of types on which Pythia-1B is better is largest), on the types where Pythia-1B is better, it

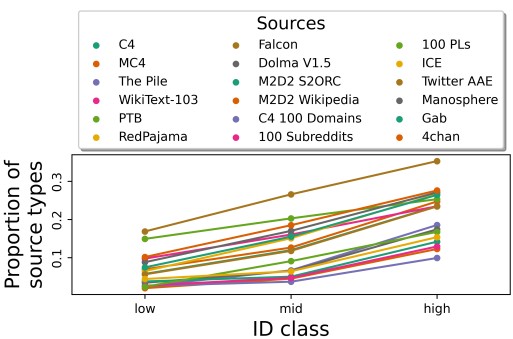

Figure 11: Proportion of types in each source for which Pythia-1B makes better predictions than Pythia-7B on the final checkpoint, as a function of type ID, $i$ (low: $i \leq 1000$; mid: $1000 < i \leq 10000$; high: $i > 10000$). The figure shows that the proportion of types for which the smaller model is better increases with type ID. Thus, while Pythia-7B is almost always better on high-frequency types (low ID), Pythia-1B is better on many low-frequency types (high ID).

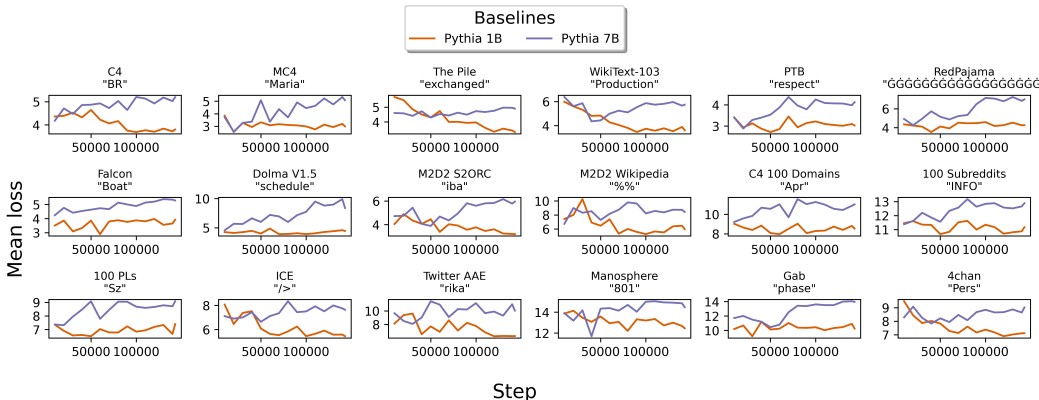

Figure 12: Training dynamics of example types for which Pythia-1B is better than Pythia-7B on the final checkpoint. We specifically show the types that, within a specific source, (i) have a minimum count of 5 and (ii) have the largest mean loss difference between Pythia-1B and Pythia-7B on the final checkpoint. We observe that sometimes Pythia-1B is better from the very beginning (e.g., *Boat* in FALCON REFINEDWEB); sometimes Pythia-1B has a constant loss while Pythia-7B is getting worse over time (e.g., *schedule* in DOLMA); sometimes Pythia-7B has a constant loss while Pythia-1B is getting better over time (e.g., *exchanged* in THE PILE); finally, sometimes Pythia-1B is decreasing its loss while Pythia-7B is increasing its loss over time (e.g., *BR* in C4).

is better on the African American domain in 77.6% of cases, and on the White aligned domain in only 71.3% of cases. In other words, there are numerous vocabulary types where the larger model performs better on the White aligned domain (as expected), and where the inverse scaling behavior only manifests itself on the African American domain.

Taken together, these results provide further evidence that reporting only aggregated perplexity values neglects more subtle dynamics on lower levels (sources, domains, vocabulary types).

## E Evaluation Data Source Details

In Table 5 we summarize each data source. All sources are existing research datasets and thus we believe our of these datasets for an evaluation benchmark is consistent with their intended use. These sources are permissively licensed and thus we are able to rehost them. Also note that we make no attempt to remove personally identifiable information (PII) beyond any filtering applied by these original datasets. As we rehost only small subsamples of these datasets and the full datasets are also

| Purpose | Source | Reference | Description |
|---|---|---|---|
| Standard language modeling benchmarks | C4 | Raffel et al. [2019] via Dodge et al. [2021] | Standard contemporary LM pretraining corpus automatically filtered from the April 2019 Common Crawl scrape |
| | MC4-EN | Chung et al. [2023] | The English language portion of a pretraining corpus automatically filtered from 71 Common Crawl scrapes |
| | WIKITEXT-103 | Merity et al. [2016] | A standard collection of verified "Good" and "Featured" articles on Wikipedia |
| | PENN TREEBANK | Marcus et al. [1999] via Nunes [2020] | Classic Wall Street Journal benchmark with linguistic structure annotations omitted |
| | REDPAJAMA | Together Computer [2023] | A publicly available reproduction of the LLaMA [Touvron et al., 2023] pretraining source mixture, combining large amounts of webscraped text with smaller curated sources |
| | FALCON REFINEDWEB | Penedo et al. [2023] | A corpus of English sampled from all Common Crawl scrapes until June 2023, more aggressively filtered and deduplicated than C4 and MC4-EN |
| | DOLMA | Soldaini et al. [2024] | A three trillion token corpus that samples sources commonly used to train LMs in order to enable open research on pretraining data |
| Fine-grained domain benchmarks | M2D2 S2ORC | Reid et al. [2022] | Papers from Semantic Scholar grouped by hierarchical academic field categories |
| | M2D2 WIKIPEDIA | Reid et al. [2022] | Wikipedia articles grouped by hierarchical categories in the Wikipedia ontology |
| | C4-100-DOMAINS | Chronopoulou et al. [2022] | Balanced samples of the top 100 URL domains in C4 as measured by page count |
| | DOLMA-100-SUBREDDITS | Soldaini et al. [2024] | Balanced samples of the top 100 subreddits by number of posts, sourced from the DOLMA Reddit subset |
| | DOLMA-100-PROGRAMMING-LANGUAGES | Kocetkov et al. [2022] via Soldaini et al. [2024] | Balanced samples of the top 100 programming languages by number of tokens, sourced from the DOLMA Stack subset |
| Communities disparities | TWITTERAAE | Blodgett et al. [2016] via Liang et al. [2022] | Balanced sets of tweets classified as African American or White aligned English |
| Fringe sources previously studied for problematic discourse | MANOSPHERE CORPUS | Ribeiro et al. [2021] | 9 forums where a set of related masculinist ideologies developed over the 2000s and 2010s |
| | GAB CORPUS | Zannettou et al. [2018] | Data from 2016-2018 from an alt-right, free-speech-oriented social media platform shown to contain more hate speech than mainstream platforms |
| | 4CHAN CORPUS | Papasavva et al. [2020] | Data from 2016-2019 from a politics subforum of an anonymity-focused forum found to contain among the highest rates of toxic content |

Table 5: Descriptions of the 16 data sources sampled to create language modeling evaluations in PALOMA. These are grouped by their purposes for inclusion (§2).

publicly available, any malicious use of these datasets would simply bypass any additional filtering we could do by using the original datasets. Also our subsampling is random and thus does not make it easier for malicious use to aggregate PII.

In the rest of this section we provide details of our use of each source and list all domains if any in each source.

**C4** Initially the pretraining corpus used by Raffel et al. [2019] and later released in Dodge et al. [2021], C4 has become one of the most commonly used pretraining corpora and is often included in more recently curated corpora. It uses a single April 2019 Common Crawl scrape to source webtext. This is filtered to remove text that is not classified as English as well as heuristics to remove text that is not natural language and a blocklist of profane keywords. We sample from the validation split of this "cleaned" corpus to measure model fit to webtext from a single temporal slice of scraping with baseline preprocessing. This source has no marked domains.

**MC4-EN** Chung et al. [2023] release a dataset with same the methods used in C4 but scale up to all Common Crawl scrapes up to August 2022 and include 107 classified languages. As the scope of the present work is the evaluation of English language models we sample only from the validation split of the English portion of the data. This allow us to measure the fit of models to scraped webtext with heterogeneous temporality. This source has no marked domains.

**WIKITEXT-103 and PENN TREEBANK** We include these two benchmarks as they have seen the most consistent evaluation on large LMs. WIKITEXT-103 [Merity et al., 2016] consists Wikipedia

articles marked "Good" and "Featured" and was used in the evaluation of GPT-2 [Radford et al., 2019], Gopher [Rae et al., 2021], and Chinchilla [Hoffmann et al., 2022]. PENN TREEBANK [Marcus et al., 1999] consists of 1989 Wall Street Journal articles originally annotated for linguistic structure. GPT-2 [Radford et al., 2019] and GPT-3 [Brown et al., 2020] omit these annotations and evaluate perplexity on the underlying text. We sample the same version of the benchmark, which is hosted by Nunes [2020]. As was standard practice at the time the benchmark is pretokenized and uncommon words are replaced with a special unknown token; we opt not to detokenize this data as we find contemporary LMs are often able to achieve comparable performance to the GPT-3 SOTA without this. These two sources have no marked domains.

**REDPAJAMA** Together Computer [2023] reproduce a pretraining corpus following the data mixture of LLaMA [Touvron et al., 2023], which combines curated sources and webscraped text similarly to THE PILE but with a much greater portion of scraped data as has become customary in recent pretraining corpora. This dataset is used to train RedPajama-INCITE [Together Computer, 2023], one of the few models with both checkpoints and data publicly available. We sample their 7 domains (see Table 6).

| arxiv, books, c4, commoncrawl, github, stackexchange, wikipedia |
| --- |

Table 6: Domains in REDPAJAMA

**FALCON REFINEDWEB** Included in the training of the Falcon models [Almazrouei et al., 2023], Penedo et al. [2023] collect a corpus of English sampled from all Common Crawl scrapes until June 2023. While we include other Common Crawl based corpora, this one has a higher duplication removal rate than previous corpora. They also claim to have more neutral filters that rely on simple interpretable heuristics and only blocklist adult content by URLs. We sample this to examine how differences in filtering scraped data influence perplexity evaluations. This source has no marked domains.

**DOLMA** Soldaini et al. [2024] curate a corpus from Common Crawl, Wikipedia, books, academic papers, code repositories, and Reddit—domains similar to those used to train most contemporary LLMs. They release the code used to collect and process this data which in combination with the corpus serve as a set of scientific artifacts to support broader participation in research on pretraining data. We sample from held out splits of each of these domains (see Table 7) to provide corresponding evaluations for these artifacts.

| books, common-crawl, pes2o, reddit_uniform, stack_uniform, wiki |
| --- |

Table 7: Domains in DOLMA

**M2D2 S2ORC** Reid et al. [2022] collect academic papers from S2ORC [Lo et al., 2020] and organize them into a two level hierarchy by academic field categories. Top-level domains, such as Computer Science, are already provided in S2ORC using top-level disciplines from the Microsoft Academic Graph [Shen et al., 2018], while subdomains are identified by a paper's arXiv category, such as the subdomain Computation and Language within Computer Science. As academic papers are a common source for pretraining and a domain for downstream use, we sample from this corpus to measure fine-grained fit to different academic disciplines. We sample both their top-level domains and lower-level subdomains, as our definition of domain accepts that domains may overlap. Also note that while the M2D2 paper only reports 106 domains and subdomains of S2ORC data, we find that there are actually 167 domains and subdomains (see Table 8) marked in their final corpus. Unfortunately the original collection concatenates together all papers, making it impossible to recover document boundaries. We resort instead to sampling a given number of tokens from the beginning of the concatenated sequences as one long pseudo-document, relying on the random shuffling of the original data before concatenation.

Art, Philosophy, astro-ph, astro-ph.CO, astro-ph.EP, astro-ph.GA, astro-ph.HE, astro-ph.IM, astro-ph.SR, astro-ph_l1, atom-ph, chem-ph, cond-mat, cond-mat.dis-nn, cond-mat.mes-hall, cond-mat.mtrl-sci, cond-mat.other, cond-mat.quant-gas, cond-mat.soft, cond-mat.stat-mech, cond-mat.str-el, cond-mat.supr-con, cond-mat_l1, cs.AI, cs.AR, cs.CC, cs.CE, cs.CG, cs.CL, cs.CR, cs.CV, cs.CY, cs.DB, cs.DC, cs.DL, cs.DM, cs.DS, cs.ET, cs.FL, cs.GL, cs.GR, cs.GT, cs.HC, cs.IR, cs.LG, cs.LO, cs.MA, cs.MM, cs.MS, cs.NA, cs.NE, cs.NI, cs.OH, cs.OS, cs.PF, cs.PL, cs.RO, cs.SC, cs.SD, cs.SE, cs.SI, cs.SY, cs_l1, econ.EM, econ.TH, econ_l1, eess.AS, eess.IV, eess.SP, eess_l1, gr-qc, hep-ex, hep-lat, hep-ph, hep-th, math.AC, math.AG, math.AP, math.AT, math.CA, math.CO, math.CT, math.CV, math.DG, math.DS, math.FA, math.GM, math.GN, math.GR, math.GT, math.HO, math.KT, math.LO, math.MG, math.NA, math.NT, math.OA, math.OC, math.PR, math.QA, math.RA, math.RT, math.SG, math.SP, math_l1, nlin.AO, nlin.CD, nlin.CG, nlin.PS, nlin.SI, nlin_l1, nucl-ex, nucl-th, physics.acc-ph, physics.ao-ph, physics.app-ph, physics.atm-clus, physics.atom-ph, physics.bio-ph, physics.chem-ph, physics.class-ph, physics.comp-ph, physics.data-an, physics.ed-ph, physics.flu-dyn, physics.gen-ph, physics.geo-ph, physics.hist-ph, physics.ins-det, physics.med-ph, physics.optics, physics.plasm-ph, physics.pop-ph, physics.soc-ph, physics.space-ph, physics_l1, plasm-ph, q-bio, q-bio.BM, q-bio.CB, q-bio.GN, q-bio.MN, q-bio.NC, q-bio.OT, q-bio.PE, q-bio.QM, q-bio.SC, q-bio.TO, q-bio_l1, q-fin.CP, q-fin.EC, q-fin.GN, q-fin.MF, q-fin.PM, q-fin.PR, q-fin.RM, q-fin.ST, q-fin.TR, q-fin_l1, quant-ph, stat.AP, stat.CO, stat.ME, stat.ML, stat.OT, stat_l1, supr-con

Table 8: Domains in M2D2 S2ORC

**M2D2 WIKIPEDIA**    Reid et al. [2022] also collect Wikipedia articles and organize them by the top two levels of hierarchy from the Wikipedia ontology. We sample from this source, as the Wikipedia ontology provides some of the largest scale human categorization of domains of text available on a data source almost always included in pretraining corpora. This time we find that their corpus contains just 49 marked domains or subdomains (see Table 9), rather than the 60 mentioned in the paper. Again the original collection concatenates articles together, so we sample a given number of tokens from the beginning of this concatenated sequence.

Culture_and_the_arts, Culture_and_the_arts__Culture_and_Humanities, Culture_and_the_arts__Games_and_Toys, Culture_and_the_arts__Mass_media, Culture_and_the_arts__Performing_arts, Culture_and_the_arts__Sports_and_Recreation, Culture_and_the_arts__The_arts_and_Entertainment, Culture_and_the_arts__Visual_arts, General_referece, General_referece__Further_research_tools_and_topics, General_referece__Reference_works, Health_and_fitness, Health_and_fitness__Exercise, Health_and_fitness__Health_science, Health_and_fitness__Human_medicine, Health_and_fitness__Nutrition, Health_and_fitness__Public_health, Health_and_fitness__Self_care, History_and_events, History_and_events__By_continent, History_and_events__By_period, History_and_events__By_region, Human_activites, Human_activites__Human_activities, Human_activites__Impact_of_human_activity, Mathematics_and_logic, Mathematics_and_logic__Fields_of_mathematics, Mathematics_and_logic__Logic, Mathematics_and_logic__Mathematics, Natural_and_physical_sciences, Natural_and_physical_sciences__Biology, Natural_and_physical_sciences__Earth_sciences, Natural_and_physical_sciences__Nature, Natural_and_physical_sciences__Physical_sciences, Philosophy_and_thinking, Philosophy_and_thinking__Philosophy, Philosophy_and_thinking__Thinking, Religion_and_belief_systems, Religion_and_belief_systems__Allah, Religion_and_belief_systems__Belief_systems, Religion_and_belief_systems__Major_beliefs_of_the_world, Society_and_social_sciences, Society_and_social_sciences__Social_sciences, Society_and_social_sciences__Society, Technology_and_applied_sciences, Technology_and_applied_sciences__Agriculture, Technology_and_applied_sciences__Computing, Technology_and_applied_sciences__Engineering, Technology_and_applied_sciences__Transport

Table 9: Domains in M2D2 WIKIPEDIA

**C4-100-DOMAINS**    Chronopoulou et al. [2022] collect C4-100-DOMAINS comprising all the text from 100 internet domains with the most pages in C4. We sample from each of the 100 domains (see Table 10) to explore the relationship between how well represented and how surprising a domain is. The original collection removes documents smaller than 200 whitespace separated tokens, leading the domain with the 3rd most pages (do5.b00kmedia.ru) to be completely empty. Only three other domains have less data than the 100 thousand tokens per split that we aim for.

100_www.ign.com, 10_www.eventbrite.com, 11_link.springer.com, 12_www.chicagotribune.com, 13_www.foxnews.com, 14_www.aljazeera.com, 15_www.dailymail.co.uk, 16_www.ncbi.nlm.nih.gov, 17_www.express.co.uk, 18_en.m.wikipedia.org, 19_www.cnet.com, 1_www.nytimes.com, 20_www.telegraph.co.uk, 21_www.theatlantic.com, 22_forums.macrumors.com, 23_www.oreilly.com, 24_www.washingtonpost.com, 25_www.zdnet.com, 26_www.foxbusiness.com, 27_www.reuters.com, 28_www.ibtimes.co.uk, 29_www.rt.com, 2_en.wikipedia.org, 30_www.prweb.com, 31_www.deviantart.com, 32_www.si.com, 33_www.bbc.com, 34_github.com, 35_nypost.com, 36_itunes.apple.com, 37_www.instructables.com, 38_www.youtube.com, 39_www.booking.com, 40_www.etsy.com, 41_www.marketwired.com, 42_sites.google.com, 43_www.baltimoresun.com, 44_www.agreatertown.com, 45_www.npr.org, 46_www.fool.com, 47_www.tripadvisor.com, 48_www.bbc.co.uk, 49_lists.w3.org, 4_www.latimes.com, 50_mashable.com, 51_disneyparksmomspanel.disney.go.com, 52_www.cnbc.com, 53_answers.sap.com, 54_homestars.com, 55_www.hindustantimes.com, 56_www.reference.com, 57_www.city-data.com, 58_medium.com, 59_app-wiringdiagram.herokuapp.com, 5_www.theguardian.com, 60_www.csmonitor.com, 61_www.adweek.com, 62_docs.microsoft.com, 63_www.yahoo.com, 64_www.thesun.co.uk, 65_www.nydailynews.com, 66_www.dailystar.co.uk, 67_fineartamerica.com, 68_www.kickstarter.com, 69_uk.reuters.com, 6_www.huffpost.com, 70_www.insiderpages.com, 71_www.inquisitr.com, 72_lists.debian.org, 73_www.straitstimes.com, 74_www.cbsnews.com, 75_simple.wikipedia.org, 76_deadline.com, 77_www.androidheadlines.com, 78_www.wired.com, 79_www.bustle.com, 7_patents.google.com, 80_premium.wpmudev.org, 81_www.librarything.com, 82_mail-archives.apache.org, 83_scholars.duke.edu, 84_www.glassdoor.com, 85_www.pcworld.com, 86_www.shutterstock.com, 87_myemail.constantcontact.com, 88_www.eventbrite.co.uk, 89_www.fastcompany.com, 8_www.businessinsider.com, 90_www.firstpost.com, 91_www.entrepreneur.com, 92_www.breitbart.com, 93_techcrunch.com, 94_www.nme.com, 95_www.ndtv.com, 96_finance.yahoo.com, 97_archives.lib.state.ma.us, 98_www.gsmarena.com, 99_www.lonelyplanet.com, 9_www.forbes.com

Table 10: Domains in C4-100-DOMAINS

**DOLMA-100-SUBREDDITS**    Using the Reddit data collected in DOLMA [Soldaini et al., 2024], we organize a new corpus of the top 100 subreddits (community forums within the messageboard) ranked by number of posts in the DOLMA data (see Table 11). In DOLMA Reddit posts are each separate

documents, without any linearization of conversational threads. Though this prevents the assessment of model fit to dialogue, it still allows evaluation across these many domains of social media text. The DOLMA Reddit data also filters out comments shorter than 500 characters and submissions (i.e., original posts) shorter than 400 characters. We sample these subreddits to capture domains as they are self-organized and self-identified by online communities.

---

00_AskReddit, 01_politics, 02_AmItheAsshole, 03_worldnews, 04_relationships, 05_relationship_advice, 06_news, 07_leagueoflegends, 08_todayilearned, 09_TwoXChromosomes, 10_personalfinance, 11_changemyview, 12_unpopularopinion, 13_movies, 14_Games, 15_nba, 16_pics, 17_gaming, 18_soccer, 19_nfl, 20_explainlikeimfive, 21_conspiracy, 22_atheism, 23_AskMen, 24_videos, 25_sex, 26_raisedbynarcissists, 27_NoStupidQuestions, 28_DestinyTheGame, 29_anime, 30_DnD, 31_ukpolitics, 32_funny, 33_europe, 34_canada, 35_Christianity, 36_SquaredCircle, 37_AskWomen, 38_legaladvice, 39_JUSTNOMIL, 40_technology, 41_IAmA, 42_wow, 43_Parenting, 44_exmormon, 45_AdviceAnimals, 46_childfree, 47_unitedkingdom, 48_ffxiv, 49_dndnext, 50_ADHD, 51_loseit, 52_asoiaf, 53_BabyBumps, 54_Advice, 55_australia, 56_CFB, 57_offmychest, 58_PublicFreakout, 59_TrueOffMyChest, 60_science, 61_magicTCG, 62_asktransgender, 63_DotA2, 64_neoliberal, 65_whowouldwin, 66_depression, 67_WTF, 68_pathofexile, 69_PoliticalDiscussion, 70_Libertarian, 71_PurplePillDebate, 72_Fitness, 73_books, 74_dogs, 75_pcmasterrace, 76_teenagers, 77_stopdrinking, 78_Overwatch, 79_television, 80_buildapc, 81_askscience, 82_programming, 83_Guildwars2, 84_cars, 85_formula1, 86_sysadmin, 87_hockey, 88_india, 89_SubredditDrama, 90_DMAcademy, 91_dating_advice, 92_Catholicism, 93_Drugs, 94_trees, 95_boardgames, 96_Conservative, 97_Futurology, 98_beyondthebump, 99_weddingplanning

Table 11: Domains in DOLMA-100-SUBREDDITS

**DOLMA-100-PROGRAMMING-LANGUAGES**  Using code repository data from THE STACK [Kocetkov et al., 2022] as it is contained in DOLMA [Soldaini et al., 2024], we collect a new corpus of balanced samples of the top one hundred programming languages by number of tokens (see Table 12). DOLMA uses an already near-deduplicated version of THE STACK, filters data related extensions (e.g., JSON and CSV) and repetitive preambles, and applies quality heuristics (e.g., removing repos with few stars). While code data differs greatly from natural language, complicating the interpretation of perplexity analysis, we nevertheless wish to add evaluations to cover this common data source for LLMs.

---

00_text, 01_markdown, 02_c, 03_php, 04_java, 05_c++, 06_python, 07_javascript, 08_html, 09_c#, 10_yaml, 11_go, 12_typescript, 13_xml, 14_css, 15_jupyter-notebook, 16_rust, 17_unity3d-asset, 18_gettext-catalog, 19_ruby, 20_vue, 21_sql, 22_swift, 23_kotlin, 24_scala, 25_scss, 26_tex, 27_dart, 28_kicad, 29_shell, 30_smali, 31_lua, 32_restructuredtext, 33_perl, 34_diff, 35_ini, 36_jsx, 37_haskell, 38_gnuplot, 39_postscript, 40_groff, 41_turtle, 42_fortran, 43_makefile, 44_mathematica, 45_pascal, 46_common-lisp, 47_gas, 48_vhdl, 49_julia, 50_edn, 51_visual-basic, 52_powershell, 53_g-code, 54_ocaml, 55_java-server-pages, 56_solidity, 57_graphviz-dot, 58_less, 59_twig, 60_asciidoc, 61_groovy, 62_llvm, 63_hcl, 64_html+erb, 65_erlang, 66_elixir, 67_eagle, 68_arduino, 69_coffeescript, 70_toml, 71_cuda, 72_nix, 73_smalltalk, 74_cmake, 75_actionscript, 76_glsl, 77_systemverilog, 78_haxe, 79_f#, 80_max, 81_objective-c++, 82_standard-ml, 83_dockerfile, 84_emacs-lisp, 85_scheme, 86_clojure, 87_handlebars, 88_smarty, 89_logos, 90_stata, 91_yacc, 92_nimrod, 93_tcl, 94_viml, 95_asp, 96_protocol-buffer, 97_r, 98_cython, 99_mediawiki

Table 12: Domains in DOLMA-100-PROGRAMMING-LANGUAGES

**TWITTERAAE**  Blodgett et al. [2016] create a pair of corpora representing African-American and White-aligned English using a statistical model with distant supervision from geolocation and demographic census statistics. We follow the reproduction of this dataset used in HELM [Liang et al., 2022], but we fix an error in loading escaped sequences of the data that, among other issues, renders emojis as literal hexadecimal bytes. Our reproduction is not able to sample the same documents, but is otherwise identical. We sample these corpora to examine disparities in performance on minoritized dialects (see Table 13).

---

AA, white

Table 13: Domains in TWITTERAAE

**MANOSPHERE CORPUS**  Ribeiro et al. [2021] curate a corpus of texts spanning 2006 to 2019 scrapped from 9 forums sharing a masculinist ideology: 8 independent message boards as well as 56 subreddits on Reddit. Using a toxicity classifier and lexicon-based misogyny metric, they find an increase in toxicity and hate over time to levels far above mainstream Reddit and comparable to 4CHAN CORPUS. We sample this corpus to measure fit to a discourse with a *specific* variety of toxicity focused on hate towards women. Moreover we intend this to exemplify how domain expertise allows the manual curation of a corpus to represent a whole discourse using known relationships between sources. The original data already linearizes the posts into a sequential thread, which we concatenate together with post authors prepended to posts. Though this datasets marks 9 domains

(see Table 14), we opt to treat this whole source as a single domain for the present analysis and thus do not perform a stratified sample of these domains.

| avfm, incels, love_shy, mgtow, pua_forum, red_pill_talk, reddit, rooshv, the_attraction |
| --- |

Table 14: Domains in MANOSPHERE CORPUS

**GAB CORPUS** Zannettou et al. [2018] scrape posts from August 2016 and January 2018 on Gab, an alt-right focused Twitter alternative founded in 2016. The platform emphasizes freedom of speech and minimal moderation, with notable users joining after being banned from mainstream social media. The authors find that GAB CORPUS measures higher than Twitter but lower than 4CHAN CORPUS on a lexicon of hate words. We sample this corpus to measure fit to low moderation social media. We treat posts as independent documents, rather than attempting to reconstruct connected subgraphs of posts replying to other posts. This source has no marked domains.

**4CHAN CORPUS** Papasavva et al. [2020] collect posts between June 2016 and November 2019 from the Politically Incorrect board (/pol/) of 4chan, a fringe imageboard emphasizing anonymity and ephemerality. Users can post content without registering, with a thread consisting of an image and message followed by a sequence comments. Threads are deleted shortly after they become inactive. As noted previously, 4CHAN CORPUS has toxicity and mysogynist hate comparable to the worst data in MANOSPHERE CORPUS and hatespeech above GAB CORPUS. We sample this corpus to measure fit to types of discourse and toxicity that can arise from anonymous posting. We concatenate posts in a thread together with post metadata prepended as a header. This source has no marked domains.

### E.1 Removed sources

Two additional sources were included in early versions of PALOMA, but were removed as access restrictions on these datasets prevent us from rehosting them. We nevertheless present their details here as we still share our findings on these datasets in this Appendix as auxiliary results not part of PALOMA.

**THE PILE** Gao et al. [2020] curate a pretraining corpus from 22 domains in one of the first large open corpora to include mostly non-webscraped text, such as archives of novels or academic papers. It is also explicitly framed as a language modeling benchmark with instructions for standardized evaluations on the validation and test sets, and several open source models have been trained on it [Wang and Komatsuzaki, 2021, Black et al., 2022, Biderman et al., 2023]. It has 22 domains (see Table 15).

| ArXiv, BookCorpus2, Books3, DM_Mathematics, Enron_Emails, EuroParl, FreeLaw, Github, Gutenberg_PG-19, HackerNews, NIH_ExPorter, OpenSubtitles, OpenWebText2, PhilPapers, Pile-CC, PubMed_Abstracts, PubMed_Central, StackExchange, USPTO_Backgrounds, Ubuntu_IRC, Wikipedia_en, YoutubeSubtitles |
| --- |

Table 15: Domains in THE PILE

**ICE** Local research teams following guidelines established in Greenbaum and Nelson [1996] collected corpora of English from Canada, East Africa (Kenya & Tanzania), Hong Kong, India, Ireland, Jamaica, Philippines, Singapore, and the USA. Each of these samples of English from around the world is further split into a written and transcribed spoken corpus, except for USA which only has written data (see Table 16). We follow HELM [Liang et al., 2022] in utilizing this corpus to measure disparate performance between these dialects. To permit comparability to HELM, we follow the same preprocessing which leaves in some XML-style tags marking phenomena such as speaker turns.

| CANADA_S_ALL, CANADA_W_ALL, EAST_AFRICA_S_ALL, EAST_AFRICA_W_ALL, HONG_KONG_S_ALL, HONG_KONG_W_ALL, IN-DIA_S_ALL, INDIA_W_ALL, IRELAND_S_ALL, IRELAND_W_ALL, JAMAICA_S_ALL, JAMAICA_W_ALL, PHILIPPINES_S_ALL, PHILIP-PINES_W_ALL, SINGAPORE_S_ALL, SINGAPORE_W_ALL, USA_W_ALL |
| --- |

Table 16: Domains in ICE

# F   Reweighting Perplexities

Even though we sample equal token counts for each domain, sometimes users of PALOMA may wish to compute a perplexity over the original distribution of domains in standard corpora such as THE PILE to compare to previous evaluations that do a uniform instead of stratified sample of these sources. We do not use such reweighted numbers in this paper, but we explain here how one might do this if desired. Instead of having to run inference twice for each source (e.g., a copy of THE PILE sampled uniformly as well as a stratified sample by domain), one can compute a perplexity with the already computed average negative log likelihood per domain $\text{NLL}_{d,c}$. Formally, for each domain $d \in D$ within a corpus $c$, consisting of a set of documents $N_{d,c} = \{t^1, \ldots, t^{|N_{d,c}|}\}$, with $\mathbf{T}(N_{d,c})$ denoting the number of tokens in that domain (i.e., $\mathbf{T}(N_{d,c}) = \sum_{t \in N_{d,c}} | \mathbf{tokenize}(t) |$) the $\text{NLL}_{d,c}$ is computed as:

$$\text{NLL}_{d,c} = -\frac{1}{\mathbf{T}(N_{d,c})} \sum_{t \in N_{d,c}} \sum_{i=1}^{|t|} \ln p(t_i | t_{<i})$$

We have $\text{NLL}_{d,c}$ where $c$ is a source in PALOMA where each domain is represented by the same number of tokens. However if we want perplexity for some other corpus $c'$ with a different distribution of domains, we can use its ratio of tokens in a domain to total tokens, $\alpha_{d,c'}$, to reweight domains:

$$\alpha_{d,c'} = \frac{\mathbf{T}(N_{d,c'})}{\sum_{d' \in D} \mathbf{T}(N_{d',c'})}$$

Now we can compute the perplexity for the domain distribution of $c'$.

$$\text{perplexity} = \exp\left(\sum_{d \in D} \alpha_{d,c'} \text{NLL}_{d,c}\right)$$

# G   Baseline Models

The 6 baseline 1B parameter models that we train employ the following architecture: 2048 maximum sequence length, 2048 model dimension, 16 layers, 16 attention heads, RoPE embedding [Su et al., 2021], SwiGLU activation [Shazeer, 2020], mixed precision, non-parametric layer normalization, and sequential model blocks for attention and feed-forward networks. We use EleutherAI's GPT NeoX tokenizer [Black et al., 2022] but add 3 additional special tokens that are used to mask PII in DOLMA. We train to 35k steps (∼150B tokens) with the following LionW optimizer [Chen et al., 2023] configurations: 2.0e-4 peak learning rate, warm-up of 2000 steps, cosine decay to 70k steps (∼300B tokens), 0.1 weight decay, and betas of 0.9 and 0.95. Note that our batch size varies slightly to accommodate two groups of baselines that were run on different hardware. The DOLMA and FALCON REFINEDWEB baselines were run with a batch size of 2112 training instances per step on 24 A100s for 9 days per model. The REDPAJAMA, THE PILE, C4, and MC4-EN baselines were run with a batch size of 2048 on 64 AMD Instinct MI250X GPUs for 2 days per model. In each case we save model checkpoints every 5k steps (∼20B tokens).

We also include baseline results from the Pythia models [Biderman et al., 2023]. These models do not conform with training guidelines (§3). They do, however, use the GPTNeoX-20B tokenizer

[Black et al., 2022] which has an identical vocabulary to our own baseline models, except lacking 3 special tokens used in DOLMA. Another similarity is that the Pythia models also have a learning rate schedule set to end at 300B tokens seen, though they train for the full 300B tokens while we train for just 150B tokens of that schedule. This permits comparison between partially trained checkpoints.

## H    Formatting and Subsampling

| | | | Evaluation Subset Tokens | | | | |
| | | | 4M | 8M | 12M | 16M | 20M | 40M |
|---|---|---|---|---|---|---|---|---|
| Train Toks | Concat | 2B | 92.23 +- 17.33 | 87.05 +- 1.82 | 86.06 +- 7.41 | 95.11 +- 26.34 | 94.91 +- 20.23 | 77.49 +- 2.34 |
| | | 26B | 21.58 +- 3.48 | 19.93 +- 2.67 | 20.24 +- 5.09 | 22.2 +- 2.24 | 22.9 +- 2.15 | 21.61 +- 2.02 |
| | | 86B | 17.94 +- 2.02 | 19.76 +- 0.79 | 20.36 +- 2.23 | 19.61 +- 1.67 | 20.25 +- 1.72 | 20.25 +- 2.43 |
| | | 286B | 16.55 +- 0.91 | 17.77 +- 1.91 | 16.7 +- 3.36 | 14.68 +- 1.86 | 17.12 +- 1.98 | 20.07 +- 3.25 |
| | Not concat | 2B | 42.57 ± 0.29 | 42.67 ± 0.14 | 42.73 ± 0.16 | 42.66 ± 0.10 | 42.69 ± 0.14 | 42.73 ± 0.09 |
| | | 26B | 21.98 ± 0.16 | 22.02 ± 0.08 | 22.04 ± 0.09 | 22.00 ± 0.04 | 22.01 ± 0.06 | 22.03 ± 0.05 |
| | | 86B | 18.52 ± 0.13 | 18.55 ± 0.07 | 18.57 ± 0.07 | 18.54 ± 0.03 | 18.55 ± 0.05 | 18.56 ± 0.04 |
| | | 286B | 16.14 ± 0.11 | 16.18 ± 0.06 | 16.19 ± 0.06 | 16.16 ± 0.03 | 16.17 ± 0.04 | 16.18 ± 0.03 |

Table 17: Average perplexity over 4 subsets of C4 validation data using Pythia 1.4B checkpoints. On top, inputs are maximum-sequence-length concatenations of random documents drawn from 4 different seeds in each cell. On bottom, random documents drawn from the same 4 seeds in all cells are evaluated separately.

We find preliminary evidence that the monotonic decrease in variability with increased evaluation or training data (see Appendix C.2.1) depends on using the non-concatenated inference input format detailed in Appendix C.2.3. In Table 17 we see that the previously observed trends break down when inputs are concatenated. Additionally, the concatenated documents are drawn from 4 random shufflings where the 4 seeds change for each cell. For comparison the bottom of the table shows results when documents are evaluated separately and with the same set of 4 random seeds for all cells. In both input formats documents that are longer than the model context window are split into separate inputs with no overlap.

We hypothesize that the trends differ between the concatenated and not concatenated formats because documents are interrupted at the start and end of concatenated instances. The location of this split will depend on the lengths of the other randomly selected documents included in the concatenation. In the non-concatenated format, documents can still be split if they exceed the maximum sequence length, but the location of the split will be the same across all random shufflings. However it is possible that other factors such as influence across document boundaries in concatenated inputs might play a role, or simply that changing the random seeds between each cell discovers more of the most unlucky, outlier seeds.

## I    Most and Least Improved Domains

In Appendix D.1 we show that improvement of LM fit when scaling is unequal from domain to domain. Differences in improvement rates can actually indicate several different training dynamics, exemplified in Figure 7. Looking at performance curves over the underlying factor of scale, helps show more specifically what is going on. Examining the domains at the extreme values of improvement rate is one way to surface interesting details of model fit. In Figure 13 we examine performance curves of the most and least improved domains with respect to number of tokens seen, $\Delta_t(\sim 20B, \sim 150B)$, and in Figure 14 we examine the most and least improved with respect to number of model parameters, $\Delta_p(85M, 805M)$ and $\Delta_p(805M, 6.4B)$.

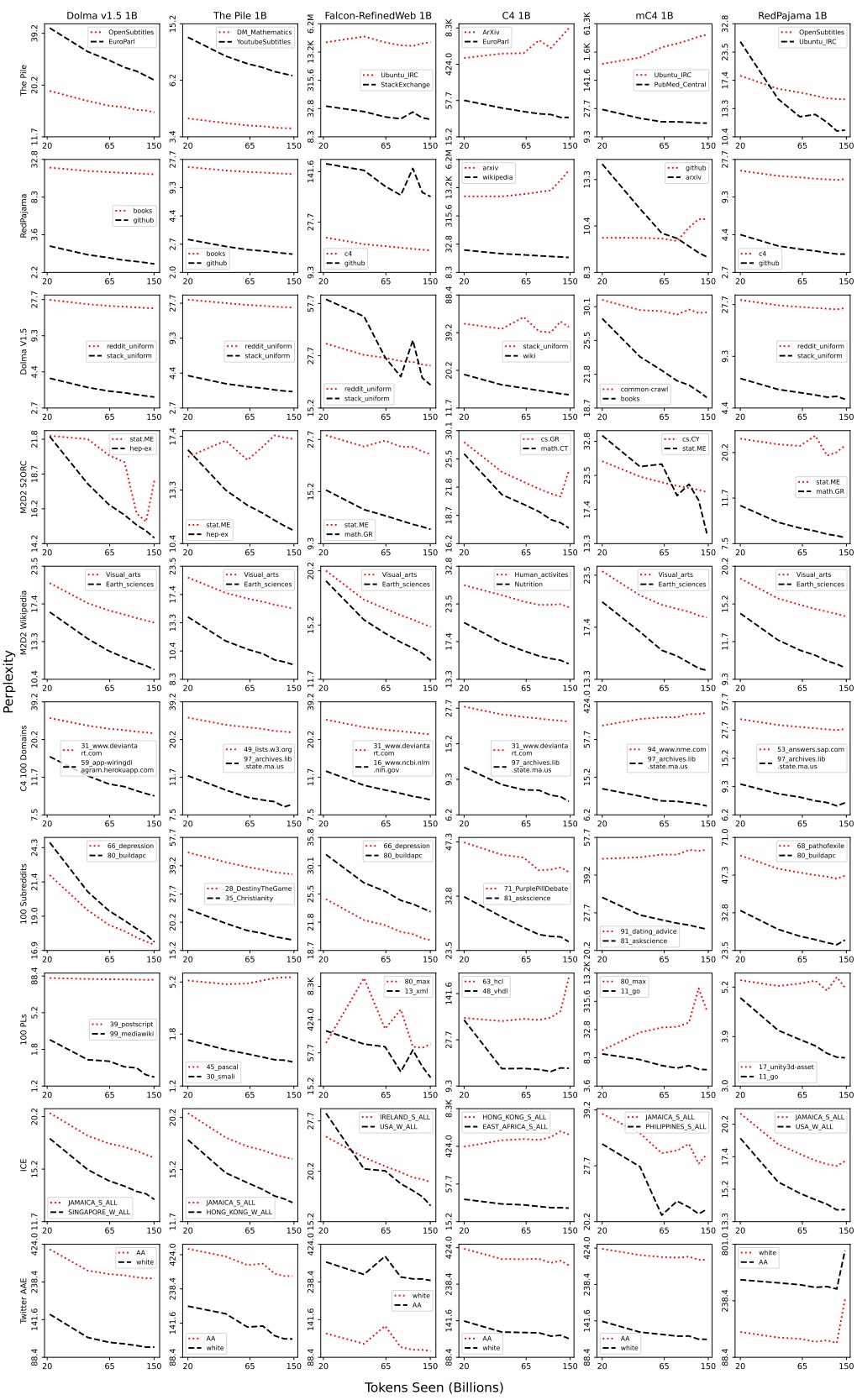

Figure 13: Perplexity curves for the most and least improved domains over an increase in tokens seen (See Appendix D.1.1). Columns are specific baseline models; rows are specific evaluation sources.

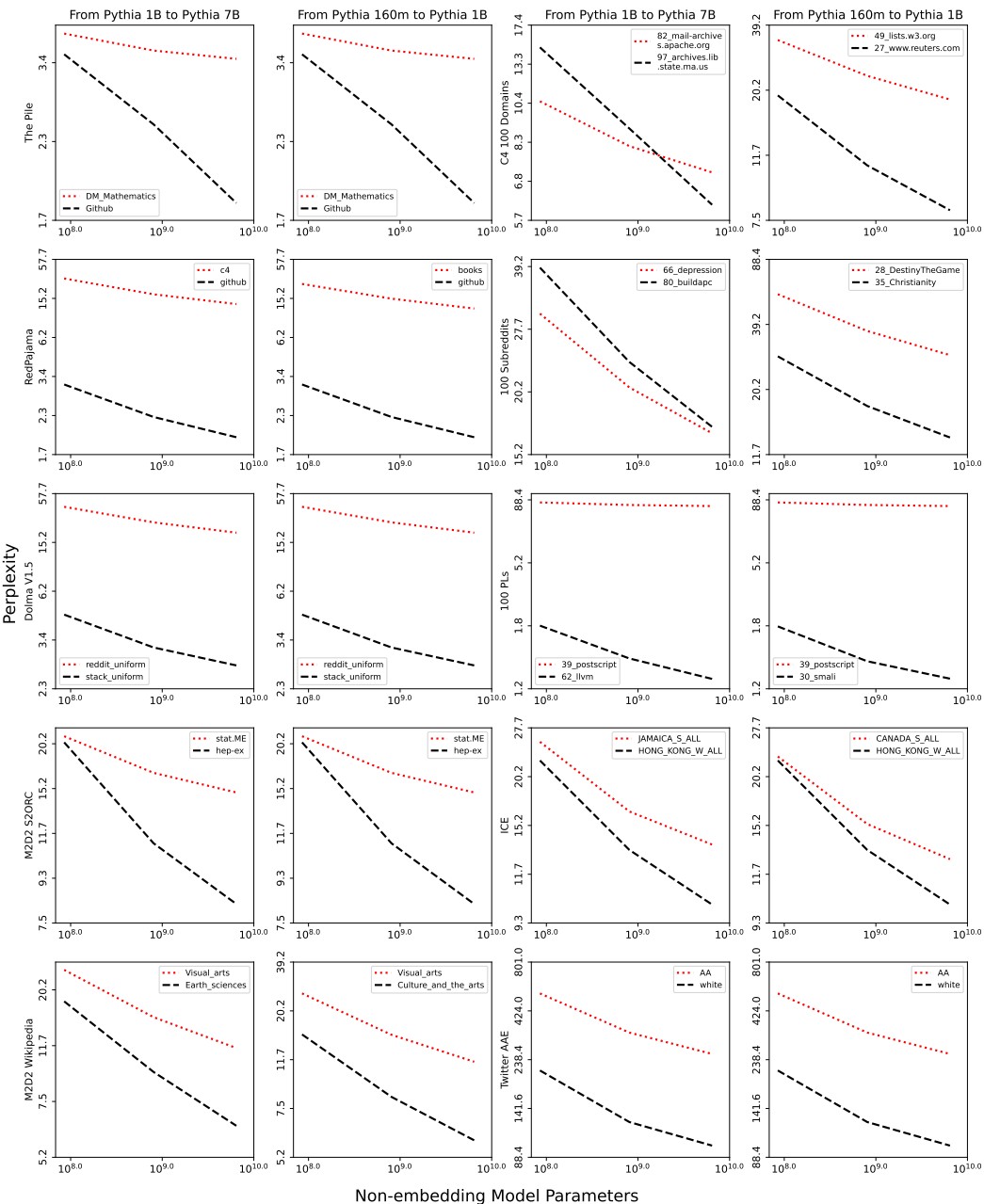

Figure 14: Perplexity curves for the most and least improved domains over an increase in model size (See Appendix D.1.2). Columns are comparisons of specific model sizes. Each row shows first one (left two subplots) and then another (right two subplots) set of evaluation sources.

