# OpenReview forum: "Paloma: A Benchmark for Evaluating Language Model Fit"
_NeurIPS.cc/2024/Datasets_and_Benchmarks_Track — NeurIPS 2024 Track Datasets and Benchmarks Poster_

### Official Review · Reviewer_5GQQ · 2024-07-19
**Valuable Benchmark for LM development, promoting a hollistic view on LM fit**

**Rating:** 7
**Confidence:** 4
**Clarity:** The paper is clearly written and well…

**Review:**

The paper is clearly written, adheres to scientific standards and present novel research. I believe the contribution to be highly significant as it presents a hollistic view on evaluating LM fit, something which has been underrepresented in literature related to evaluation of LMs. In addition, the work also presents insightful case studies that provide further evidence for the value of the dataset, and tools are released to decontaminate training data.

**Pros**
- Covers a wide range of domains and allows users to evaluate LMs more hollistically.
- The value of the Benchmark and conclusions that can be drawn from it is shown with experiments.
- Well written and structured paper.
- Release of tools to decontaminate training data.

**Cons**
- It can be more thouroughly motivated why perplexity is a useful metric. Along these lines, I believe the paper would benefit from a clearer discussion of the limitations of Pereplexity.
- While the Benchmark covers a wide area of domains, users of the benchmarks need to decontaminate possibly terrabytes of training data. The paper would benefit from emphasizing the importance of decontamination more.

**Strengths:**

- promotes a more hollistic view on evaluating LMs with respect to language model fit, covering 565 English and code domains.
- provides a meticulous categorization of datasets into different sources and subdomains.
- provides detailed guidelines on how to evaluate perplexity and its intricacies.
- show interesting novel insights, including that validation loss on the C4 dataset gives an incomplete picture of LM fit and that models trained only on web data, have high perplexity on other domains.

**Additional Feedback:**

-

**Correctness:**

I believe the claims in the paper to be correct and the dataset has been constructed in a sound way. It is also worth emphasizing that the authors took great care to present the evaluation metrics and its intricacies in a sound and transparent way.

**Documentation:**

The paper, together with the released code provides sufficient documentation.

**Ethics:**

Ethical considerations are discussed at a satisfactory level.

**Limitations:**

The limitations of this work and its societal impact are to a large extent adequately emphasized. As mentioned above, I believe the limitations of perplexity can be emphasized more clearly.

**Opportunities For Improvement:**

The main area where I see room for improvement is on making a stronger case on why/when perplexity is a good metric to evaluate LMs and on decontamination. I think providing a better motivation along these lines can make the contribution of the paper stronger.

**Correlation Analysis**
While the paper presents a detailed view on the language modelling objective, it could further benefit from making a connection with performance on downstream tasks. I would find it particularly insightful to see an analysis on how perplexity on each of the 16 sources in Paloma correlates with a set of high-signal tasks.

**Limitations of Perplexity**
Second, I think the paper would further benefit from more thoroughly highlighting the limitations and opportunities of using perplexity to evaluate LMs. On the limitation side, it is not clear whether perplexity as a whole, and on which domain in particular, is predictive for different downstream tasks. This ties with my previous comment on a correlation analysis. On the opportunity side, perplexity as a continuous metric allows smaller/undertrained models to be compared against each other. This can also be highlighted more prominently.

**Decontamination**
Since the benchmark covers broad domains, it also presents the challenge of possibly decontaminating Terrabytes of training data against the benchmark. To provide more guidance for users, I think the paper would benefit from more detailed results/discussions on the importance of decontamination and how results might be distorted when training data is not decontaminated.

**Relation To Prior Work:**

Most related work is clearly discussed. However, I would like to see the paper being more thoroughly situated and compared to other benchmarks (on a qualitative level). E.g., a section on how Paloma relates to other means of evaluating language models such as HELM [1],  evaluation

**Summary And Contributions:**

This paper introduces a benchmark designed to evaluate language models (LMs) with respect to the language modelling objective across 500+ language and code domains collected from various openly accessible sources. The core contribution of the paper is the case for need of evaluating language model fit on diverse domains, rather than on a single domain (e.g., on wikipedia) as is often the practice. Next to the dataset itself, 6 1B LMs are released, together with code to reproduce the results and decontaminate future training datasets.

---

> ### Author Response · Authors · 2024-08-15
> **Response to 5GQQ19**
>
> We thank the reviewer for their attentive and encouraging review! We are glad that the reviewer **appreciated our writing**, the **holism of our analysis**, and **our standardized decontamination approach for removing test leakage**.
>
> >**Correlation Analysis**
>
> Please see our new correlation results and discussion in the separate response comment titled “Correlation of downstream tasks and fine-grained perplexity.” We particularly appreciate the reviewer’s concrete suggestion for correlation of the 16 sources to downstream evaluations. We will include this analysis in our revision.
>
> >**Limitations of Perplexity**
>
>
> We see fine-grained perplexity evals as an important complement to existing evaluations that comes with different limitations than downstream benchmarking. One of our key findings, as highlighted by reviewer Camw25, is the limitation that perplexity is dominated by the top 5 percent of the vocabulary, which suggests that only examining perplexity aggregated over all vocabulary types may obscure differences between fine-grained domains. Following reviewer kc9d31’s suggestion we will add a stand-alone limitations section to consolidate the limitations currently discussed throughout the paper. And regarding the limitations of what our 6 pretraining experiments can show about the association between perplexity and downstream performance we hope that our discussion in the previously mentioned separate comment also addresses your concern.
>
> >**Decontamination**
>
>
> We welcome the reviewer’s encouragement to further emphasize the importance of decontaminating benchmark leakage into pretraining data. The confounding effect of contamination is a key concern for us and the first of our guidelines for Perplexity evaluations done right (section 3). We provide further details of our decontamination approach in Appendix B.1.1 including:
> 1. A review of literature investigating the impact of contamination. In particular we note that results in [Lee et al. (2022)](https://arxiv.org/abs/2107.06499) address perplexity specifically and show that models underestimate perplexity on evaluation documents with near duplicates in the training corpus by several points relative to models with those duplicate training documents removed.
> 2. We explain how the Bloom Filter approach which we employ scales more easily to terrabytes of pretraining data than previous deduplication approaches like minHash and Suffix arrays.
> 3. We discuss our choice of paragraph-level granularity for matching and our choice of heuristics for minimum matching size.
> 4. We discuss how we remove contamination once it is identified, and present results on document removal rates for each of the corpora we use to train our 6 baselines.
> 5. We note limitations to our approach including its non-applicability to code and that removing full documents based on any partial contamination will bias towards removing longer documents.
>
> In our revision we will follow the reviewer’s suggestion and place more emphasis on motivating decontamination, especially in discussing what is known from controlled ablations of decontamination techniques, such as  [Lee et al. (2022)](https://arxiv.org/abs/2107.06499)’s, about how much evaluations can be distorted.
>
> >**Relation to Prior Work**
>
> In Table 2 and the paragraph starting line 189, we situate Paloma among other fine-grained perplexity datasets, including the language modeling tasks in HELM. Further low level comparisons are made to previous benchmark implementations in Appendix B regarding deduplication (line 699), use of BPB (line 840), and input format (859). In our revision we will include more high level comparison to non-perplexity benchmarks in our discussion of correlations to downstream tasks (as mentioned above).

---

### Official Review · Reviewer_zXDY · 2024-07-23

**Rating:** 7
**Confidence:** 4
**Correctness:** Yes
**Clarity:** Yes

**Review:**

### Advantages

1.  PALOMA's inclusion of 546 distinct domains allows for a nuanced understanding of LM performance, moving beyond traditional single-dataset evaluations. This comprehensive approach enables researchers to identify specific areas where models excel or falter, fostering targeted improvements.

2.  The paper provides a rigorous framework for evaluating perplexity across different models, ensuring that comparisons are fair and reproducible. The controlled parameters (e.g., sample size, vocabulary) enhance the reliability of the findings, which is crucial for scientific discourse.

3. Case studies illuminate how models pretrained on limited or homogeneous datasets, such as Common Crawl, may exhibit significant shortcomings in fit across varied domains. This insight encourages the exploration of richer, more diverse pretraining strategies, ultimately driving better model performance.


### Disadvantages

1. While PALOMA focuses on English and code data, this narrow scope may limit its applicability to multilingual contexts or domains with different linguistic characteristics.

2. The benchmarking process requires significant computational resources for training and evaluating the LMs, which may not be accessible to all researchers.

Overall, PALOMA represents a significant advancement in the evaluation of language models, offering a robust framework that highlights the importance of domain diversity in model training. I stand positive.

**Strengths:**

See above

**Additional Feedback:**

None

**Documentation:**

Yes

**Ethics:**

No ethical concerns

**Limitations:**

Yes

**Opportunities For Improvement:**

See above

**Relation To Prior Work:**

Yes

**Summary And Contributions:**

The paper introduces PALOMA , a benchmark designed to evaluate LM fit across diverse domains. Different with the traditional reliance on perplexity measured on single, monolithic datasets, PALOMA includes two new datasets: one from the top 100 subreddits and another from the most common programming languages, totaling 546 domains. The benchmark provides six baseline 1B parameter LMs pretrained on various corpora, alongside controlled experimental guidelines. The authors demonstrate the utility of PALOMA through case studies that reveal findings about LM fit and the impact of pretraining data diversity.

---

> ### Author Response · Authors · 2024-08-15
> **Response to zXDY23**
>
> We thank the reviewer for their thoughtful, positive review! We are glad that the reviewer **appreciated comprehensiveness of our benchmarks**, our **rigorous standardized evaluation framework**, and our **case study on the impacts of heterogeneous pretraining data sourcing**.
>
> >**Focus on English and code (D1)**.
>
> We completely agree that our focus on English and code data is a major limitation of Paloma. We select this scope as unfortunately most current LMs (and pretraining corpora) also focus on these types of data. However we intend that our “rigorous framework for evaluating perplexity” (as noted by the reviewer) is generalizable to the construction of new perplexity benchmarks. We strongly encourage future work to explore using this framework to construct multilingual perplexity benchmarks. We will add this suggestion (currently on line 101) to the consolidated limitations section recommended by reviewer kc9d31 which we will add in our revisions.
>
> >**Cost of pretraining benchmarking (D2)**
>
> We aim to minimize the inference costs of our benchmark by subsetting based on observed metric variance over varying subset sizes (our 3rd guideline in section 3), resulting in a benchmark that is comparable or smaller in number of tokens to existing validation sets for C4 or Pile. As a pretraining benchmark, the training costs will necessarily be larger. This problem is our key motivation (line 43) for introducing a common set of standards for comparability, so that pretraining experiments that are undertaken can be comparable with as many other experiments as possible.

---

### Official Review · Reviewer_Camw · 2024-07-25
**a more fine-grained domain perplexity measurement for LLM pretraining**

**Rating:** 6
**Confidence:** 3
**Correctness:** yes

**Review:**

A proper LLM evaluation is a crucial problem that can significantly help people's understanding of LLMs and also boost the development in the field. Past perplexity based evaluation is based on the held-out data from the training datasource domain. This paper presents an evaluation that separates hundreds of domains to enable more clarified performance evaluation.

There're many places i find confusing:
1. Figure 3: "For each source with at least 10 domains, each point visualizes perplexity on a single domain for a fully trained model". What do you mean here? It's very confusing. Also, what's the X-axis ? The tokens seen in units of Billions? Why is the median perpelexity increasing?
2. The behavior anaylysis on the incredibly high perplexity for Red-pajama. I don't get what conclusions it gives.
3. Also it mentions at the beginning that LLM fit is not a good measure for domains when talking about red-pajama -- then does it mean the perplexity loss is not a good measurement? If so,  it defeats the purpose of this paper itself. But if not, what is a better measurement or a set of analysis you should do? It's unclear.
4. How is your newly prosposed evaluation dataset curation different from the Dolma's held out data?

**Strengths:**

perplexity measurement is easy; a good evaluation benchmark based on it is definitely helpful for the comunnity.

**Additional Feedback:**

no

**Clarity:**

This papper could use a lot of improvement for its writing (points already listed in my "review" section)

**Documentation:**

yes

**Opportunities For Improvement:**

Besides a bunch of points listed in the above "Review" section that I need clarifications, I think there's a core issue that could make this paper significantly better --  How does each domain's perplexity score affects common downstream task?

It's nice to have a fine-grained domain perplexity measurement, but it's unclear what they indicate so far. Admittedly what we can do is always limited, but it'd be much helpful to perform some downstream tasks (as often reported in any LLM model release) and analyze how they correlate with the domain perplexity measurement.

**Relation To Prior Work:**

yes

**Summary And Contributions:**

This paper provides a more diverse and finegrained LLM perplexity-based evaluation dataset curation. Based on it, the author ablates some common pretraining dataset and showed their performance gap, highlighting the insufficiency of Common-Crawl only dataset and the imbalance of the improvement in different domains.  Also the paper finds some interesting phenonmena such as the most loss reduction come from the top 5 percent of the vocabulary.

---

> ### Author Response · Authors · 2024-08-15
> **Response to Camw25 (part 1)**
>
> We thank the reviewer for their detailed and helpful review! We are glad that the reviewer sees the **potential for our perplexity benchmark to be helpful for the community** and especially happy that the reviewer was interested in our favorite **finding that loss reduction is dominated by the top 5 percent of vocabulary**.
>
> >**About Figure 3 (Q1)**
>
> The x axis in this figure is not tokens but actually the rank of the domain as ordered by median perplexity among the baseline models on that domain. For example the domain at the 0th tick on the x axis has the lowest median perplexity as measured over the 6 baseline models. Points above tick are perplexity scores for that specific domain for each of the 6 baseline models. We arrange the domains by their median perplexity over the baselines, as this order gives some sense of the intrinsic difficulty of a domain (line 274). We propose clarifying this novel visualization by revising the x axis label to “Rank of Evaluation Domain, as Ordered by Median Perplexity over Models.”
>
> >**Redpajama perplexity anomaly example (Q2)**
>
> The reviewer may be referring to the high perplexities of our redpajama (RPJ) baseline model, which we attribute to a training loss spike in the model on line 269. Specifically this model suffers training instability shortly before its final checkpoint and we observe that in some evaluations, the final checkpoint also spikes in perplexity. Or the reviewer may be referring to line 259 where we show high perplexity on one domain in RPJ eval data (arxiv) as one example along a programming language domain in the Dolma code source. We observe that all such anomalies occur for models trained on Common Crawl only data and that the evaluation domains all sample nonnatural language such as code. We hypothesize that these anomalous perplexities may occur when nonnatural strings are not seen at training due to single-source pretraining corpora only having one set of cleaning heuristics which may filter out some strings entirely.
>
> >**What models do learn vs should learn (Q3)**
>
> The reviewer is likely referring to the paragraph starting on line 89, “better fit may not reflect what is valued by all the communities who produce language in these domains [[Diaz and Madaio, 2023](https://arxiv.org/abs/2307.03201)].“ This is not saying that LM fit is not a good measure of a model's ability to perform as a language model in a given domain. This is clarifying that what the specific humans producing the language being evaluated consider to be good performance cannot align with any one metric. This is simply because of the anthropological fact that human values are extremely diverse and beyond the scope of one metric. In response we say “instead of relying on LM fit to represent alignment to a domain’s human salient features, we examine anomalies in domain fit to deepen understanding of language modeling dynamics and illuminate gaps in existing approaches to evaluation,” following an approach to interpretability outlined by [Holtzman et al. [2023]](http://arxiv.org/pdf/2308.00189) and [McCoy et al. [2023]](http://arxiv.org/pdf/2309.13638). In our revision we should more simply say, perplexity shows us what a model is learning as opposed to what the model should be learning. We then give an example in which academic paper domains from different sources (RPJ and Dolma in this instance) can have dramatically different perplexity for a given model. The point of this example is that seemingly overlapping domains can still differ in human unintuitive ways such as unintentional differences in filtering and formatting, and that examining fine-grained perplexity reveals these differences.
>
> >**Paloma is not Dolma’s held out set (Q4)**
>
>
> Dolma’s held out set is one of the sources we sample to create Paloma, *along with many others*. Specifically we take stratified samples of Dolma’s held out set to create the “Dolma” source in Paloma. The “Dolma-100-Subreddits” and “Dolma-100-Programming-Languages” sources are also constructed from the Dolma held out set by taking stratified samples of the top 100 subreddits/PLs as marked by included metadata. In sum 3 of 16 of our sources sample a small portion of Dolma, but Paloma is not an official held out set of Dolma.
>
> >**Downstream Correlation Analysis**
>
> Please see our new correlation results and discussion in the separate response comment titled “Correlation of downstream tasks and fine-grained perplexity.” We particularly appreciate how the reviewer notes that there are limitations to what can be known about this topic from the 6 pretraining experiments we have conducted and we endeavor to clarify what we can see from these experiments. We will include this analysis in our revision.

---

> > ### Author Response · Authors · 2024-08-15
> > **Response to Camw25 (part 2)**
> >
> > >**Limitations**
> >
> >
> > We see fine-grained perplexity evals as an important complement to existing evaluations that comes with different limitations than downstream benchmarking. One of our key findings, as highlighted by reviewer Camw25, is the limitation that perplexity is dominated by the top 5 percent of the vocabulary, which suggests that only examining perplexity aggregated over all vocabulary types may obscure differences between fine-grained domains. Following reviewer kc9d31’s suggestion we will add a stand-alone limitations section to consolidate the limitations currently discussed throughout the paper.
> >
> > >**Clarity**
> >
> > We appreciate the reviewers clarifying questions, which will help us refine our language to avoid misunderstandings in the camera-ready.

---

### Official Review · Reviewer_kc9d · 2024-08-01
**An interesting benchmark and a nice contribution**

**Rating:** 8
**Confidence:** 3
**Correctness:** See "Review".
**Clarity:** Yes

**Review:**

**Strength**

Contributions of this work are solid, timely, and relevant. Evaluation of LMs, especially for pre-training, is a crucial topic currently attracting much attention and drawing many debates. Additional well-crafted evaluation benchmarks will provide viable tools for researchers and developers alike to better understand the mechanisms behind LM pre-training.

Compared to many evaluation benchmarks that focus on task performance (e.g., LM evaluation harness, MMLU, etc.), this work proposes to extend the evaluation from the other perspective–perplexity.  A number of works have claimed perplexity on the pre-training distribution (e.g., held-out split for C4/The Pile, etc.), which is a good indicator for tracking pre-training progress and can be used as a proxy for evaluating LM utility.  This paper delves into the assumption that is so often taken for granted and provides a much more fine-grained analysis and original findings.

**Issues**

My questions are also about these novel findings. Existing papers seem to unanimously agree that perplexity on C4, despite C4 only consisting of web text, strongly correlates with aggregated performance on many tasks. Some paper further finds that training the model prioritizing web text, such as C4/CommonCrawl, generally leads to advantageous task performance and the model trained only on C4/CommonCrawl, could achieve strong performance on downstream tasks comparable to models trained on best-tuned domain weights.

Yet, this paper shows models trained on C4 often score high perplexity on many domains while models trained on Dolma often achieves lowest perplexity. For models pre-trained on RedPajama, perplexity on certain domains will even flare up after ~100B training tokens. Is this because this benchmark focuses more on the domains from the Dolma dataset?

Does the perplexity evaluation correlate with the models' task performance? It seems crucial to me to also evaluate these models on downstream tasks and analyze how aggregated perplexity/per-domain perplexity correlates with aggregated task performance/per-category task performance. It could make this benchmark more impactful.

**Strengths:**

See "Review".

**Additional Feedback:**

See "Review".

**Documentation:**

Yes

**Limitations:**

Recommend discussing limitations in a separate subsection/paragraph.

**Opportunities For Improvement:**

See "Review".

**Relation To Prior Work:**

Yes

**Summary And Contributions:**

This paper investigates the important problem of perplexity-based evaluation for LM training. The paper claims that current efforts mostly evaluate the model based on monolithic data held out from training and report this perplexity, yet this relies on a strong assumption that perplexity on one distribution extrapolates to others, which is not necessarily true. Instead, this work proposes a new dataset for evaluating LMs, aggregating 546 English and code domains with two originally proposed subsets. The paper also provides 6 baseline 1B LMs controlled to provide fair comparisons about the utility of common pretraining corpa. Case studies conducted in the paper find models pretrained without data beyond Common Crawl exhibit certain gaps in LM fit to many domains.

---

> ### Author Response · Authors · 2024-08-15
> **Response to kc9d31**
>
> We thank the reviewer for their insightful review! We are very glad that the reviewer **appreciates the importance for perplexity evaluation** beyond monolithic held out sets and **exploring further into the assumption that perplexity from one distribution extrapolates to others**.
>
> >**Relationship of our finding about heterogeneous training stabilizing perplexity to previous work on web text scaling**
>
>
> The reviewer notes that most scaling literature uses a single validation distribution (rather than fine-grained domains as in Paloma) to measure the relation of loss to scale and proxy downstream performance. In our case study in Appendix C.1 we find that most domains consistently exhibit lower loss at larger scales though the coefficients of these relationships vary by domain and model. For the purpose of observing the general phenomenon that LM loss reduces with scale, it seems that most validation distributions will indeed do fine. However, to make concrete predictions of downstream task performance based on loss, our finding of widely varying scaling coefficients suggests care must be taken to fit this relationship for each pair of validation and training distributions. This is concurrently corroborated by [Gadre et al 2024](https://arxiv.org/abs/2403.08540) who apply the traditional scaling laws paradigm to the relationship between loss and downstream tasks over several coarse grained validation and training distributions and find that the fit of these scaling laws does indeed vary. As for work finding that web text alone can outperform heterogeneous sourced corpora with respect to downstream tasks (e.g., [Li et al 2024](https://arxiv.org/abs/2406.11794)), our finding that training on common crawl alone can lead to unstable perplexity metrics is orthogonal but possibly complementary as stable perplexity evaluation is helpful for investigations of the relationship of loss to downstream performance, further discussion of which is provided below.
>
> >**Why are the Dolma baseline’s perplexities often better than RedPajama and C4 baselines**
>
> The reviewer wonders if good fit derives from overlap of Paloma with Dolma. While the fact that Paloma does sample Dolma may contribute to this, Paloma also samples RedPajama and C4. Instead, our hypothesis (Line 240) is that the distribution aggregated by Paloma is more similar to heterogeneous corpora such as Dolma, Pile, and RPJ because of our stratified sampling. Regarding the high perplexities of our Redpajama (RPJ) baseline model on its final checkpoint, we attribute this to a training loss spike in the model (line 269). Specifically this model suffers training instability shortly before its final checkpoint and we observe that in some evaluations this particular checkpoint also spikes in perplexity.
>
> >**Downstream Correlation Analysis**
>
> Please see our new correlation results and discussion in the separate response comment titled “Correlation of downstream tasks and fine-grained perplexity.” We particularly appreciate the reviewer’s interest in how the per-domain perplexities available in Paloma may correlate with downstream tasks. We will include analysis from our response in our revision.
>
> >**Limitations**
>
>
> We see fine-grained perplexity evals as an important complement to existing evaluations that comes with different limitations than downstream benchmarking. One of our key findings, as highlighted by reviewer Camw25, is the limitation that perplexity is dominated by the top 5 percent of the vocabulary, which suggests that only examining perplexity aggregated over all vocabulary types may obscure differences between fine-grained domains. Per the reviewers suggestion, we will add a stand-alone limitations section to consolidate the limitations currently discussed throughout the paper.

---

### Author Response · Authors · 2024-08-15
**Correlation of downstream tasks and fine-grained perplexity**

Three of the reviewers expressed interest in the correlation between Paloma perplexity metrics and downstream tasks. In Table R1 below we provide the Spearman’s rank correlation between the ranking of our 6 baseline models’ final checkpoints by each of the 16 Paloma sources and by each of the 8 downstream evaluations used in OLMo ([Groeneveld et al 2024](https://arxiv.org/abs/2402.00838)).

Table R1: Values greater than abs(0.5) are bolded for emphasis.

|               |   c4-en |   mc4-en |   wikitext |   ptb |   redpajama |   falcon-rw |   dolma |   m2d2 s2orc |   m2d2 wikipedia |   c4 **100** domains |   **100** subreddits |   **100** PLs |   twitterAAE |   4chan |   manosphere |   gab |
|:--------------|--------:|---------:|-----------:|------:|------------:|------------:|--------:|-------------:|-----------------:|-----------------:|-----------------:|----------:|-------------:|--------:|-------------:|------:|
| arc_challenge |   -0.38 |     0.20 |       **0.67** |  0.49 |        **0.52** |        **0.55** |    0.38 |         0.46 |             **0.78** |            -0.23 |             0.23 |      0.23 |        -0.06 |    0.38 |         0.38 |  0.20 |
| arc_easy      |   -0.26 |     0.49 |       0.37 |  **0.71** |        0.43 |        0.31 |    0.49 |         0.31 |             **0.60** |            -0.37 |             0.14 |      **0.54** |        -0.31 |    0.49 |         0.49 |  0.03 |
| boolq         |   **-0.60** |    -0.09 |      -0.09 | -0.43 |        0.20 |       **-0.94** |   -0.31 |        -0.14 |            -0.20 |            **-0.66** |            **-0.54** |     -0.03 |        **-0.60** |   -0.31 |        -0.31 | -0.09 |
| hellaswag     |    **0.77** |    -0.03 |      **-0.83** | -0.49 |       **-0.94** |        0.14 |   **-0.54** |        **-0.71** |            **-0.77** |             **0.83** |            -0.20 |     **-0.66** |         0.37 |   **-0.54** |        **-0.54** | -0.49 |
| openbookqa    |    0.49 |    **-0.83** |      -0.03 | -0.37 |       -0.20 |        0.03 |   -0.26 |        -0.09 |            -0.14 |             **0.60** |             0.09 |     -0.43 |         0.20 |   -0.26 |        -0.26 | -0.14 |
| piqa          |    0.41 |     0.23 |      **-0.58** | -0.12 |       -0.49 |       -0.12 |   -0.06 |        -0.29 |            **-0.64** |             0.23 |             0.06 |      0.03 |         0.35 |   -0.06 |        -0.06 |  0.03 |
| sciq          |    0.23 |    -0.41 |       0.41 |  0.12 |        0.17 |        0.46 |    0.35 |         0.49 |             0.17 |             0.32 |             **0.67** |      0.12 |         **0.72** |    0.35 |         0.35 |  **0.61** |
| winogrande    |    0.03 |    **-0.61** |       **0.61** |  0.26 |        0.46 |        0.26 |    0.20 |         0.38 |             **0.67** |             0.12 |             0.20 |      0.06 |        -0.23 |    0.20 |         0.20 | -0.06 |

These results provide some indication of relationships between specific perplexity evals and downstream tasks, such as M2D2 wikipedia and arc_challenge or hellaswag and c4-en. Most importantly, it is apparent that no single perplexity evaluation correlates well with all downstream tasks, suggesting the importance of evaluating across a range of diverse perplexity evaluations rather than a single monolithic validation loss.

However we caution against reading too far into these correlations without further pretraining experiments across a greater range of compute scales and data mixes. For our set of 6 pretraining experiments, the correlation of rankings by the same *downstream* tasks between adjacent model checkpoints is only a moderate 0.513 when averaged over tasks and checkpoint pairs. This suggests that the differences in downstream performance between these mixes at this scale are not stably discernible on their own regardless of correlation to perplexity. We hope that users of our benchmark will create controlled pretraining experiments at larger scales with more distinct data mixes whose *downstream* rankings are more consistently discernible. On line 313 we note that enabling research on the relationship of perplexity and downstream tasks is one of the motivations of our benchmark. Following the reviewer’s suggestions, we will include this analysis and discussion of limitations to what we can know about downstream performance from our 6 pretraining experiments in our revision to aid future work in using Paloma to explore this topic.

---

### Decision · Program_Chairs · 2024-09-26

**Decision:**

Accept (Poster)

**Comment:**

The reviewers have reached a consensus to recommend the acceptance of this paper. All reviewers acknowledge its significant contribution to the community, mainly through its multi-domain evaluation of language models using perplexity measures. While some reviewers noted minor weaknesses, the AC believes that the paper's strengths far outweigh these issues, which can be addressed in a revised version. The AC agrees with the reviewers' positive views and recommends accepting the paper.